# The impact of land use on non-native species incidence and number in local assemblages worldwide

Daijun Liu [1] ✉, Philipp Semenchuk [1,2], Franz Essl[1], Bernd Lenzner [1], Dietmar Moser[1], Tim M. Blackburn [3,4], Phillip Cassey [5], Dino Biancolini [6,27], César Capinha[7,8], Wayne Dawson [9], Ellie E. Dyer [10,11], Benoit Guénard[12], Evan P. Economo[13,14], Holger Kreft [15,16], Jan Pergl [17], Petr Pyšek[17,18], Mark van Kleunen [19,20], Wolfgang Nentwig[21], Carlo Rondinini [6], Hanno Seebens[22], Patrick Weigelt [15,16,23], Marten Winter [24], Andy Purvis [25,26] & Stefan Dullinger [1]

While the regional distribution of non-native species is increasingly well documented for some taxa, global analyses of non-native species in local assemblages are still missing. Here, we use a worldwide collection of assemblages from five taxa - ants, birds, mammals, spiders and vascular plants - to assess whether the incidence, frequency and proportions of naturalised non-native species depend on type and intensity of land use. In plants, assemblages of primary vegetation are least invaded. In the other taxa, primary vegetation is among the least invaded land-use types, but one or several other types have equally low levels of occurrence, frequency and proportions of non-native species. High land use intensity is associated with higher non-native incidence and frequency in primary vegetation, while intensity effects are inconsistent for other land-use types. These findings highlight the potential dual role of unused primary vegetation in preserving native biodiversity and in conferring resistance against biological invasions.

The Anthropocene biodiversity crisis is driven by various facets of human activity such as direct exploitation of organisms, transformation of pristine to modified ecosystems, environmental pollution, alteration of the Earth's climate and human translocation of species beyond their native distributions[1,2]. These drivers of biodiversity change are expected to interact with each other[3–7]. However, much of the research linking drivers to biodiversity change has either focused on individual drivers[3], or on the interaction between land use (henceforth abbreviated as LU) and climate change[8,9].

Human LU – mainly land conversion and subsequent management for crop cultivation and livestock raising – has been identified as the strongest driver of biodiversity loss in terrestrial ecosystems[3,6,9] by destroying, degrading and fragmenting species' habitats across at least three quarters of the earth's ice-free land mass[10]. However, the conversion of natural ecosystems or the management of converted ones does not affect all species equally, as some species may tolerate the novel environmental conditions while others may not[11]. In particular, non-native species (i.e., species deliberately or accidentally introduced outside of their native ranges by human activity) are often among those that profit from human disturbance of native ecosystems[12–14]. Land-use change may thus promote the naturalisation and spread of non-native species[12,15,16], while their accumulation in local assemblages may mask LU-driven loss of native biodiversity[17,18]. The possible interactions between LU change and biological invasions have long been hypothesised and discussed[19–21], but a global-scale analysis of empirical data on how LU change and biological invasions interact at the scale of local assemblages has not yet been performed.

A key question in this context is how LU affects non-native species incidence and richness in local assemblages. Available studies on this topic are biased towards temperate regions and demonstrate considerable variability across ecosystems, LU-types, and taxonomic groups[14,22]. For example, data from Europe demonstrated that non-native plant species tend to colonise highly disturbed and converted habitats[12,23]. However, there are large differences between agricultural or urban ecosystems, both of which often are heavily invaded by non-native species, and grasslands or forests, which have far fewer non-native plants even when used intensively in Europe and also in North America[23–25]. In Europe, the association of non-native species with LU may also differ markedly among different taxonomic groups[22]. While high numbers of non-native plants and insects are commonly observed in intensively used ecosystems, non-native birds and mammals appear more evenly distributed between natural and human-modified environments. This finding echoes earlier studies which suggested that non-native plants may depend more on human disturbance for successful establishment than do non-native tetrapods[13]. Whether these patterns hold true beyond these regions and taxa remains largely unknown.

Here, we investigate the occurrence of non-native species in local assemblages and how these occurrences are related to LU by combining a global dataset of local assemblages in different LU-types (i.e., the PREDICTS database[26,27]) with data on the regional distribution of naturalised non-native species from five taxonomic groups: ants, birds, mammals, spiders and vascular plants. Assemblage data represent lists of species recorded at local scales (mean linear extent of sampling area = 60 m, see Methods and the reference[26] for details) in six LU-types (Primary vegetation, Secondary vegetation, Plantation, Pasture, Cropland and Urban) used by humans at three LU-intensity levels (e.g. Minimal, Light and Intense). We combined these different data sources by identifying species in the PREDICTS assemblages as non-native if they were listed as naturalised in the respective region in the species distribution databases. We then analysed the relationship between the type and intensity of human LU and non-native species incidence (i.e. whether there is at least one non-native species in an assemblage), number and proportion in the assemblage. We hypothesised (1) that primary vegetation is generally less invaded than ecosystems modified by humans, and (2) that the incidence and number of non-native species increases with the intensity of LU. However, based on previous results from Europe[22], we also expected (3) that LU promotes non-native species establishment in assemblages of different taxonomic groups to a different extent, with plants being the most responsive to LU change. We found that primary vegetation is generally among the least invaded LU types. High LU-intensity is associated with higher non-native incidence and frequency in primary vegetation, while intensity effects are inconsistent for other land-use types. Our findings highlight the importance of conserving and restoring pristine ecosytsems as they can help rescuing native biodiversity and constrain the establishment of non-native species.

## Results

Of the 26,114 local assemblages compiled in PREDICTS, 11,713 report species lists of the five taxonomic groups studied (ants (407), birds (4925), mammals (1147), spiders (773), and vascular plants (4461)). Of these 11,713 assemblages, 20.9% (2451) contained at least one non-native species (Fig. 1a; Supplementary Table 4). The percentage of assemblages with at least one non-native species was highest for vascular plants (31.5%), followed by ants (27.8%), mammals (25.5%), spiders (23.5%) and birds (9.3%) (Fig. 1c–g and Supplementary Fig. 1). The majority of assemblages containing non-natives had only one non-native species (53.1% across all taxa, and 77.9%, 67.6%, 87.4% and 65.4% for ants, birds, mammals and spiders; respectively Fig. 1b–f). Assemblages with more than one non-native species were most frequent for vascular plants (19.7%), but considerably rarer in the animal taxa analysed (ants: 6.1%, birds: 3%, mammals: 3.2%, spiders: 8.2%).

### Non-native incidence

Non-native species were more likely to occur in most combinations of LU-type and -intensity than in 'Primary vegetation under Minimal use' (Fig. 2a and Supplementary Table 5), with odds ratios ranging between 3.8 ('Primary vegetation under Intense use') and 27 ('Cropland under Intense use'). Non-native incidence was particularly likely in intensively used cropland and urban areas (in the latter, each assemblage had at least one non-native species, which led to modelling problems and this factor combination was therefore removed from the model, see Methods). When using LU-type as the only predictor in a model across all taxonomic groups taken together (Fig. 2b and Supplementary Table 6), all LU-types had significantly higher odds ratios of non-native incidence than primary vegetation. However, analysing the data separately for each taxonomic group demonstrated taxon-specific differences in the responses to LU-type (Fig. 2b–g and Supplementary Table 6), with the contrast between primary vegetation and all other types being consistently significant only for vascular plants. For all other taxa, there was always at least one LU-type that did not differ significantly from primary vegetation in the odds of non-native species incidence. The identity of these LU-types was different across groups. Only plantations had consistently higher non-native incidence in all groups where this LU-type could be included in the model (all but spiders).

When using LU-intensity as the only predictor in a model across all taxonomic groups, intensity levels 'Minimal' and 'Light' did not differ in the likelihood of non-native incidence, but assemblages under intense use had higher odds of harbouring a non-native species (Supplementary Fig. 2b). In the full model with both LU-type, LU-intensity and their interaction as predictors, the effect of intensity was not consistent across LU-types (Fig. 2a). While non-native incidence increased with intensity of usage in primary vegetation, the relationship was variable in the other LU-types.

We tested our results for sensitivity (1) to biases in the spatial distribution of assemblages (Supplementary Figs. 5a and 6a–f and Supplementary Tables 7 and 8), (2) to the fact that the sampling area used for collecting species lists of assemblages varied considerably across the studies compiled (Supplementary Table 9) and (3) to effects of some assemblages being located on an island versus a continent in PREDICTS (Supplementary Figs. 7a and 8a–f and Supplementary Tables 10 and 11). We found that reported patterns were qualitatively robust to all these possible sources of bias, except for alien incidence being higher in plantations under light use when islands were excluded from the analyses (Supplementary Figs. 5–8).

### Non-native number and proportion

Across all five taxonomic groups, non-native species numbers in assemblages containing at least one non-native were lowest in 'Primary vegetation under Minimal use' (Fig. 3a and Supplementary Table 12), with an estimated average of 1.1 non-native species. When using LU-type as the only predictor in a model across taxonomic groups, all LU-types had significantly higher non-native species numbers than primary vegetation (Fig. 4a and Supplementary Table 13). Similar to non-native incidence, this contrast was clearest in the case of vascular plants (Fig. 4b), which had significantly more non-native species than in primary vegetation in all other LU-types. For birds and spiders, only some of the LU-types had higher non-native species numbers than primary vegetation, especially urban areas for birds, and pastures for spiders. For ants, no differences among LU-types were detected. Mammal numbers were not analysed separately since the vast majority of assemblages (87.4%) contained only one non-native species.

Across all taxonomic groups, a model with LU-intensity as the single predictor did not detect any significant effect on non-native species numbers (Supplementary Fig. 3). However, the full model with LU-type, LU-intensity and their interaction as predictors (Fig. 3a) demonstrated that differences among LU-intensity levels did occur,

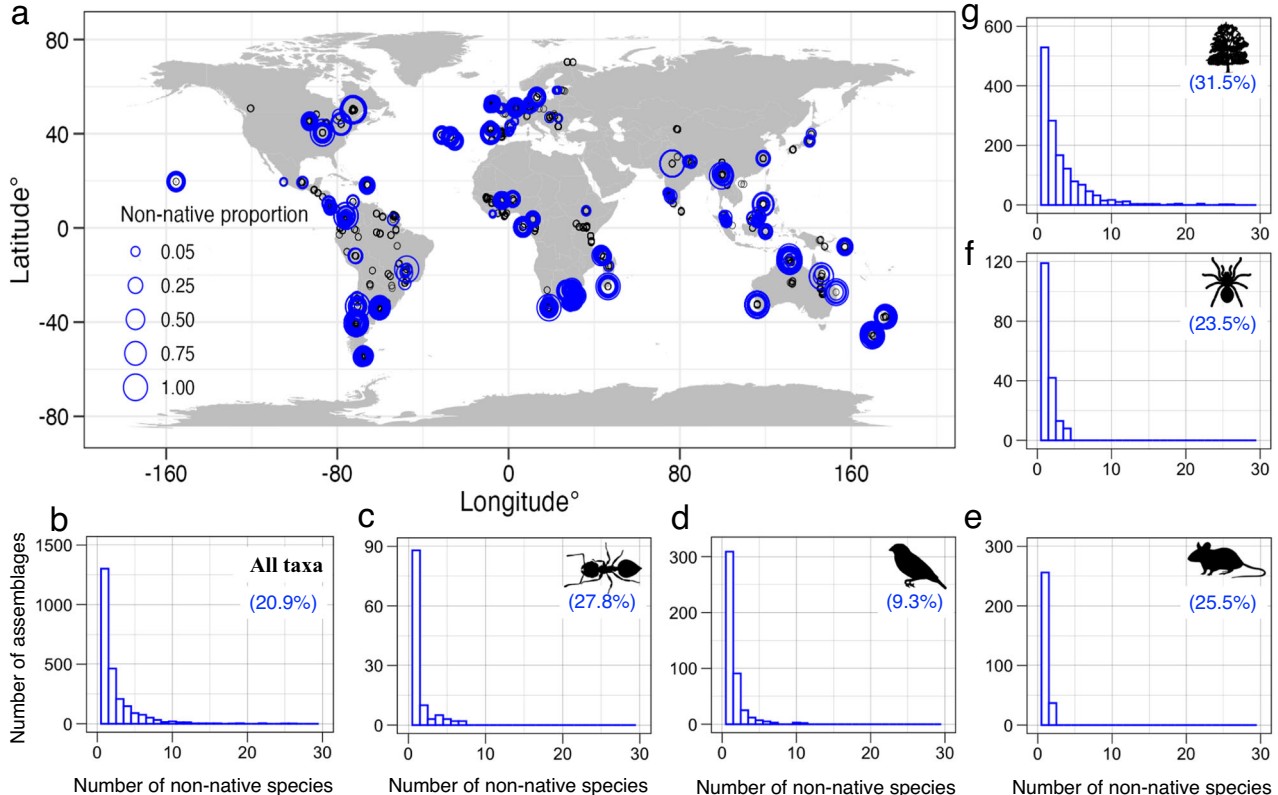

**Fig. 1 | Geographic distribution of local assemblages and histograms of the number of local assemblages with at least one non-native species. a** distribution of assemblages with at least one non-native species (blue circles) and sites without non-native species (black circles) for all five focal taxonomic groups. The different sizes of blue circles represent non-native proportions (the number of non-native species relative to all species) in local assemblages. The subplots (**b–g**) only refer to assemblages containing at least one non-native species. **b** histogram of the number of assemblages with different numbers of non-native species across all five taxonomic groups (overall); **c–g** the same histograms separated by taxonomic groups: ants, birds, mammals, spiders and vascular plants, respectively. Percentages in the subplots (**b–g**) indicate the percentage of assemblages with at least one non-native species compared to all assemblages in the dataset. Silhouette illustrations for the taxa are from PhyloPic (http://phylopic.org), contributed by various authors under public domain license.

but were limited to certain LU-types, especially primary vegetation, where higher intensity led to higher non-native species numbers (from 1.1 species under minimal to 2.1 species under intense use). The contrary was found for plantation forest and cropland, where non-native numbers were highest under minimal use.

Across all five taxonomic groups, the proportions of non-native species in local assemblages with at least one non-native showed patterns broadly similar to the number of non-native species, but with differences in some details such as the invasion status in secondary vegetation and plantation under different intensity of use (Fig. 3b, Supplementary Fig. 4 and Supplementary Table 14). When using LU-type as the only predictor in a model across taxonomic groups, non-native proportions were higher than in primary vegetation (0.27) in all types except pastures (Fig. 4c and Supplementary Table 15). As for non-native incidence and numbers, this pattern was clearer in the case of vascular plants, where non-native proportions were higher than in primary vegetation in all LU-types except pastures, and peaked in urban areas. By contrast, non-native proportions did not differ among LU-types in case of mammals and spiders. Proportions of non-native ants were more variable among LU-types than their numbers, with particularly high values in plantations (0.72). Birds had their highest non-native proportions in cropland and urban areas (0.18 and 0.2; respectively), but those were significantly different from secondary vegetation only, not from primary vegetation.

LU-intensity had weaker effects on non-native proportions than on non-native numbers, with no differences among intensity levels across groups and LU-types (Supplementary Fig. 4) and only few significant contrasts within the LU-types (Fig. 3b).

As in the case of species incidence, sensitivity analyses suggested that the results for species numbers and proportions were qualitatively robust against uneven sampling across regions (Supplementary Figs. 5b, c and 6g–j and Supplementary Tables 7 and 8), variation in the spatial extent of the local assemblage samples (Supplementary Table 9) and the location of assemblages on an island or a continent in PREDICTS (Supplementary Figs. 6b, c and 8g–j and Supplementary Tables 10 and 11). The one exception was that proportions of non-native species were lower in urban environments under intense use if islands were excluded from the analysis.

## Discussion
While the regional distribution and richness of non-native species have been increasingly documented over the last decade[14], a global analysis of non-native incidence and frequency in local assemblages has been missing to date. The data and analyses presented here demonstrate that non-native species have already colonised many assemblages. As expected, human usage of ecosystems tends to facilitate the encroachment of non-native species into local assemblages, although this effect was not detectable for all types of usage in all taxonomic groups. Considering incidence, number and proportions of non-native species in combination, differences between primary vegetation and human-used ecosystems appear most pronounced in vascular plants and least pronounced in mammals and spiders. However, an important caveat with the latter conclusion is that these two groups had low sample sizes in some LU-types. Contrary to our expectation, we found that higher LU-intensity affects non-native incidence and numbers only in primary vegetation, but has no consistent effects in other LU-types.

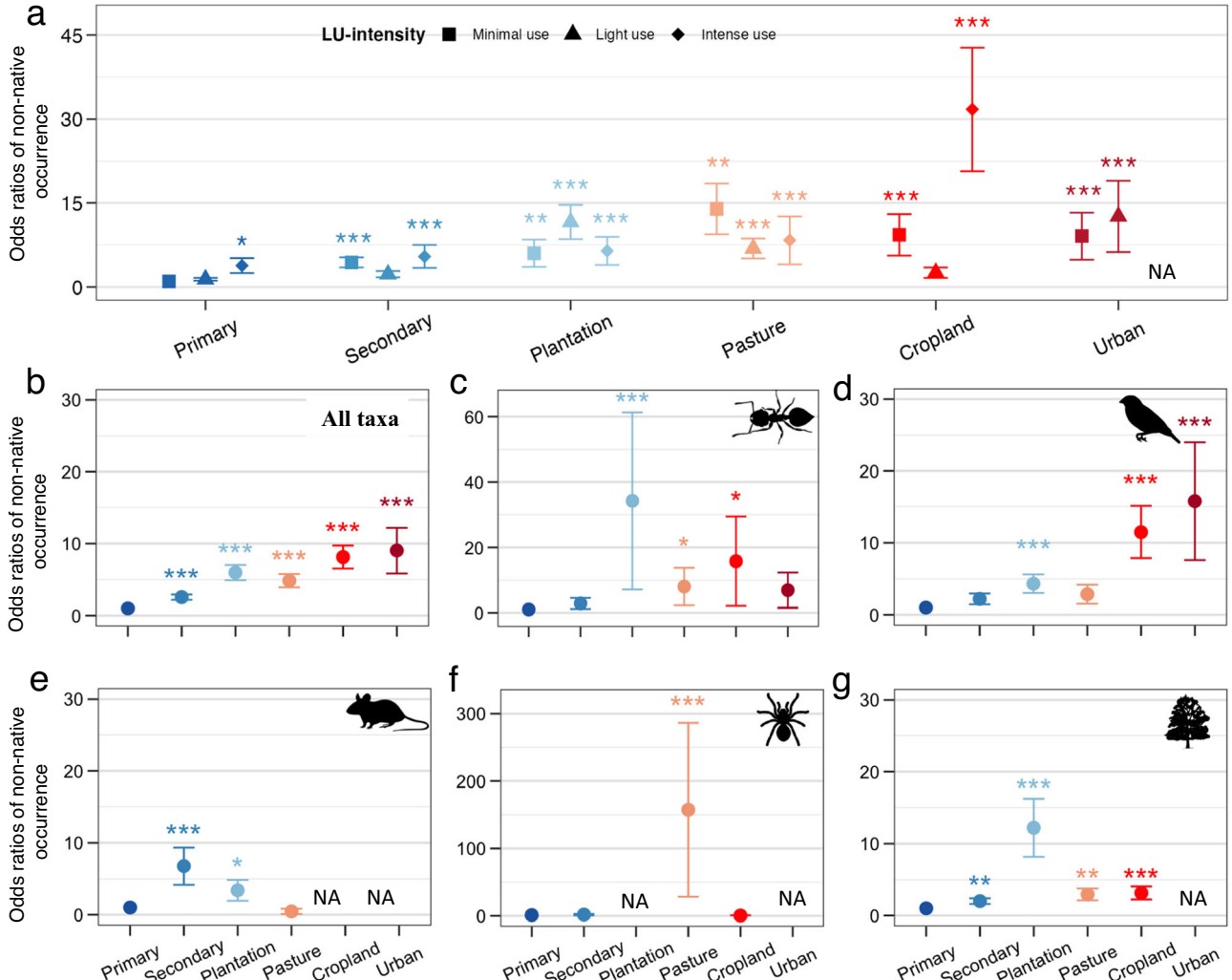

**Fig. 2 | Odds ratios of non-native species occurrence in local assemblages in response to LU-type and LU-intensity. a** a logistic generalised linear mixed effects model with LU-type, LU-intensity and their interaction as predictors ($n = 11,693$); **b**–**g** the effect of LU-type modelled across all taxa (**b**) ($n = 11,713$) and for each taxon separately (**c**–**g**) ($n = 407, 3978, 1114, 762$ and $4453$; respectively). All $p$-values are based on two-sided tests. Odds ratios from contrasts to the reference levels ('Primary under Minimal use' (**a**) or 'Primary vegetation' (**b**–**g**)) are back-transformed from the logarithmic scale of the models, respectively. Shown are mean odds ratios (dots) and standard errors (bars). Asterisks indicate significant differences from the reference levels ($p$ values: *<0.05; **<0.01 and ***<0.001). The NA in **a** and **f** indicates intense use in urban areas and spiders in plantation, which were removed from the analysis since all of the assemblages had non-native species (leading to total separation in the models, see "Methods" section for details). The NA in **e**, **f** and **g** indicates mammals, spiders and vascular plants in urban areas, and mammals in cropland, which could not be included in the analysis due to low sample size. Silhouette illustrations for the taxa are from PhyloPic (http://phylopic.org), contributed by various authors under public domain license.

We emphasise that our analysis only accounted for naturalised non-native species, while some of the assemblages may have contained non-native species not yet considered naturalised in the focal region. Since non-native species invasion is an ongoing phenomenon showing no signs of decline[7], non-native incidence, numbers and proportions calculated here are most likely conservative estimates.

**One fifth of local assemblages contained non-native species**

The percentage of local assemblages invaded by at least one non-native species is uneven across groups, with plant assemblages apparently being the most and bird assemblages the least invaded. However, the total number of species naturalised outside their native range is much higher in plants than in the other groups (Supplementary Table 2). A higher incidence of non-native plants in local assemblages would thus also result from neutral community assembly processes. Given that there are almost 50 times more non-native plants than non-native birds, mammals, spider or ants (Supplementary Table 2), the incidence of vascular plants might even appear low, and those of some other

groups high in relative terms. However, whether there really are non-random differences in non-native incidence among taxonomic groups would need a more thorough analysis. The PREDICTS data were not designed and are hence difficult to use for such an analysis, mainly because of a number of taxonomic, environmental and geographical sampling biases. For example, assemblages from primary vegetation are most frequent in the PREDICTS database (Supplementary Table 16), which might explain why vascular plants – the group most sensitive to LU change in our study – might even appear under-represented in local assemblages. In addition, residence times of non-native species are often short, i.e. they have been introduced some decades or, at most, few centuries ago[7], and their spread across local assemblages might hence be still ongoing, especially in less mobile taxa such as plants[28,29], resulting in a local-scale invasion debt[30]. The apparent over-representation of mammals, ants and spiders, by contrast, might be attributable to the fact that differences in LU have less impact on their establishment probability, they have been more ubiquitously introduced, or that a subset of particularly successful invaders has

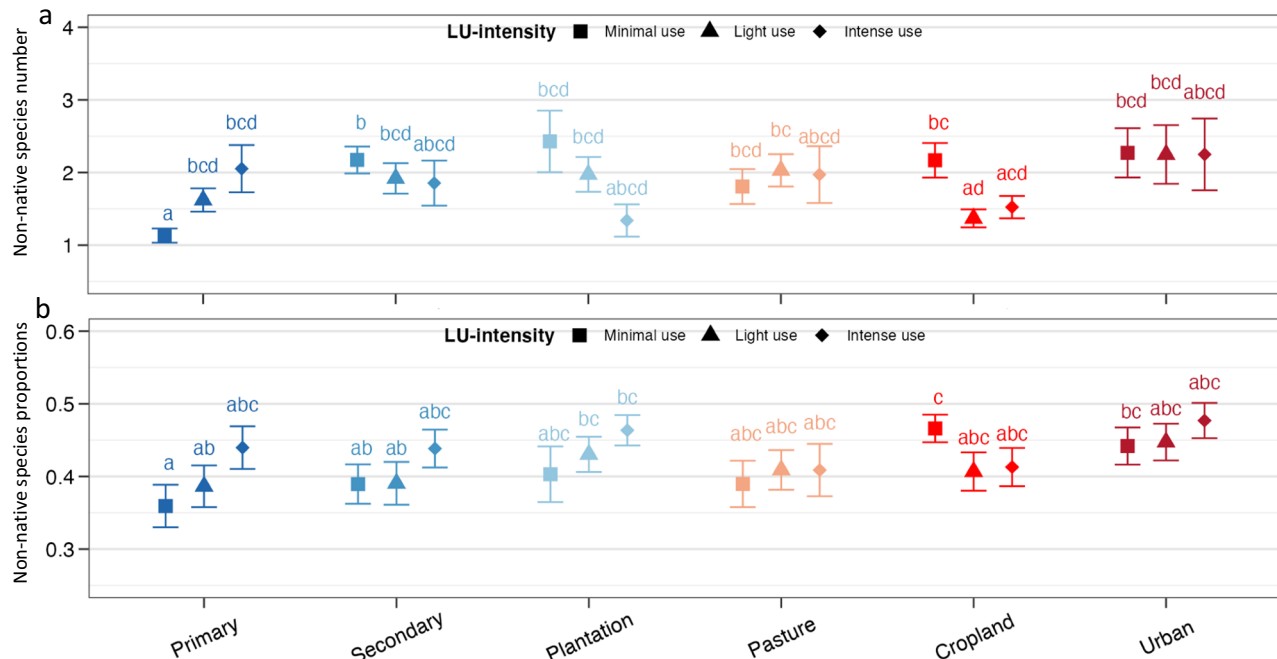

**Fig. 3 | Number of non-native species and their proportion in local assemblages in response to LU-type, LU-intensity and their interaction, analysed across all five taxa.** The values in **a** and **b** are back-transformed estimates (means and standard errors) from a generalised linear mixed effects model (GLMM) with an assumed Poisson distribution (**a**) or beta distribution (**b**) (*n* = 2314). All p-values are based on two-sided tests. The letters indicate statistical significance (*p* < 0.05) of all pairwise comparisons among the means of LU-type and LU-intensity levels.

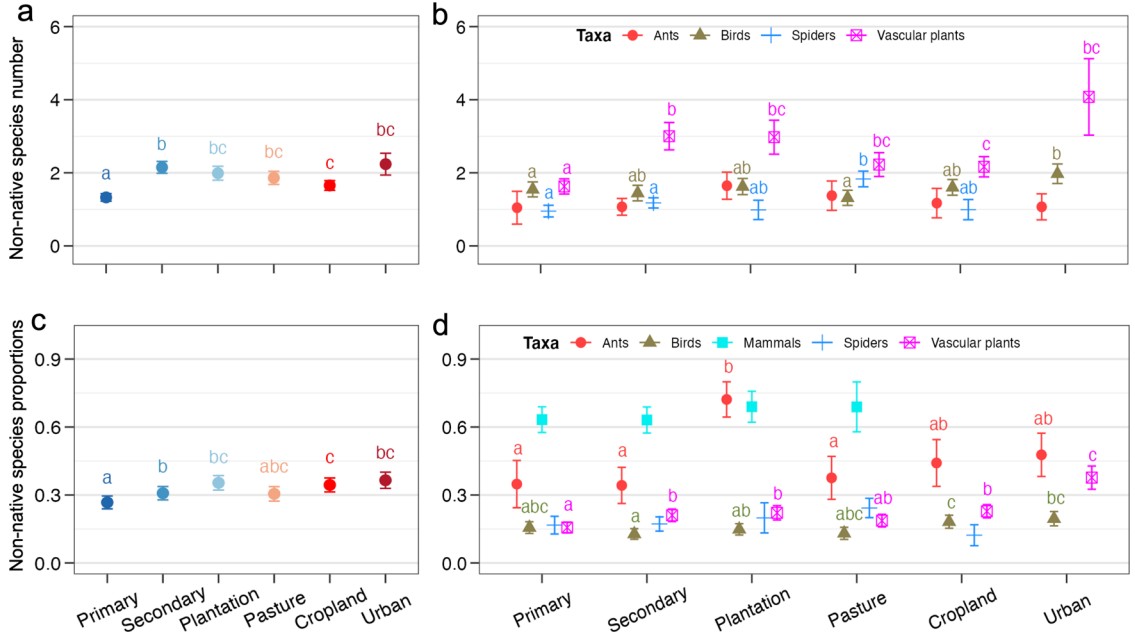

**Fig. 4 | Numbers and proportions of non-native species among all species in local assemblages containing at least one non-native species in dependence on LU-type.** Panels **a** and **c** present models fit across all five focal taxonomic groups (across all taxa) (*n* = 2450). Panels **b** and **d** models fitted for each taxon separately (*n* = 113, 457, 292, 182 and 1406; respectively). The statistical model test used are two-sided. The different letters indicate significant differences (*p* < 0.05) for all pairwise comparisons among LU-types. If there were no significant differences (Mammals and Spiders in **d**), letters were omitted. Values are back-transformed estimates (the means and standard errors) from generalised linear mixed models (GLMM) with assumed Poisson (**a**, **b**) or beta distributions (**c**, **d**). Numbers of non-native mammal species were not analysed separately since the majority of assemblages (87.4%) contained only one non-native species. Mammals in cropland and urban areas (**d**), and spiders in urban areas were not included in the model due to a lack of data (**b**, **d**).

rapidly spread across many local assemblages. Indeed, non-native mammals are peculiar in that they have been frequently introduced intentionally for a range of purposes[31,32], while ants and spiders have been dispersed a lot by the global trade of commodities[33].

**Effects of land use**

Primary vegetation is the least invaded LU-type across all groups considered. This is true for non-native incidence, the number of non-native species and also, albeit less clearly, for the proportion of

non-native species among all species. This finding is generally in line with the well-documented propensity of non-native plants to invade human disturbed conditions[23], and their comparatively low incidence in natural and semi-natural vegetation, in particular natural forests[24]. There are a number of potential explanations for these patterns, including a tendency of naturalised non-native plants to show a ruderal life history strategy[34], the often more stable and lower level of resources in primary vegetation[35,36], and a generally lower level of propagule pressure in the remote remnants of pristine ecosystems[37]. For other taxonomic groups, the evidence for a bias of non-native colonisation against primary vegetation has been less conclusive so far[22,38]. In Europe, non-native vertebrates, in particular, have been reported to be relatively more common in forests than non-native plants and non-native insects[16,22]. Our results partly corroborate these earlier findings in that the lower representation of non-native species in primary vegetation is more pronounced in plants than in all other taxa analysed. The kind of LU-type most amenable to non-native establishment appears to vary across groups, and results are also less consistent across incidence, numbers and proportions of non-native species than in the case of vascular plants. The idea that any kind of human LU facilitates non-native species is hence not fully supported by our data and may be linked to a taxonomic bias of invasion research towards plants and birds[16,39,40]. On the other hand, primary vegetation did not show higher non-native species incidence, number or proportions than any other LU-type in any of the taxonomic groups. Our data thus at least corroborate that natural or near-natural ecosystems are not particularly vulnerable to non-native species colonisation by any of the five taxa.

The invasibility of LU-types may differ across taxa for several reasons. For vascular plants, ecological opportunities arising from fluctuating resource availability typical of human-used ecosystems have long been considered to promote invasions[41]. Whether this aspect is similarly important for heterotrophic organisms, which use a broader and more differentiated array of resources, is unclear. Differences in introduction pathways and associated habitat requirements of introduced species may also play a role. The predominant pathway of vascular plant introduction is horticulture[42], creating a bias towards species growing in open or semi-open vegetation, which prevail in gardens and parks as well as in pastures, croplands and urban areas. By contrast, many non-native vertebrates have been introduced for hunting purposes or as pets[31], without a similar bias, or even a tendency towards species thriving in natural environments[22,32]. Non-native ants and spiders are usually introduced unintentionally as stowaways of traded commodities, partly timber or other forest products (e.g. fruits), and their habitat requirements may hence also be less biased than those of plants. Third, the taxonomic groups considered differ in their intrinsic mobility, with plants probably being least mobile[29]. Given that primary vegetation is often the LU-type most distant from points of non-native species introduction, plants may simply be slower in colonising primary vegetation than species from the other groups. Our results would thus document differences in the level by which species from different taxonomic groups have already realised their potential ranges in the non-native regions. Lastly, sampling bias in PREDICTS may also contribute to apparent differences in LU 'affinities' among groups, especially in cropland and urban areas (Supplementary Table 16). Non-native birds and vascular plants, the two groups documented best in the PREDICTS data, were found to be particularly frequent in these LU-types. In the case of vascular plants, this pattern is well established for Europe[12,22,23,43] and North America[44], although our data suggest that, globally, grasslands and plantations may be similarly invaded. Besides human disturbance, high propagule pressure is considered a main driver of this pattern, especially in urban areas which usually are hubs of species introduction[14]. For birds, human-dominated landscapes have also been reported to be hotspots of non-native species richness[16,45–47], including agricultural areas in Europe[22], or intensively used areas, both croplands and plantations, in

the tropics[48]. By contrast, non-native mammal, ant and spider assemblages are much less well documented in PREDICTS. As a consequence, the representation of non-native species of these groups in croplands and urban areas either could not be evaluated at all, or there is less statistical power for these groups compared to birds or vascular plants. We hence assume that for the invertebrate groups, in particular, the representation of non-native species in human-used ecosystems is underestimated in our analysis. This may also explain contradictions to other studies that have, for example, reported high frequencies of non-native ant species in intensively used pastures and plantations of non-native trees such as oil palms[49]. Also, non-native spiders are thought generally to have a synanthropic distribution pattern[50].

Besides differences in the type of usage, the PREDICTS data also suggest some dependence of non-native species on the intensity of human land management. Across all species and LU-types, intensively used ecosystems show higher non-native incidence. However, this difference is mainly driven by the situation in primary vegetation, where non-native numbers rise with rising intensity. The latter finding probably reflects that human management can alter important ecosystem properties without converting an ecosystem into a completely different one, such as forest into grassland, e.g. by selective logging or introduction of non-native trees in forest ecosystems. The effect of intense management on the invasibility of primary vegetation likely has similar reasons as the effect of converting primary vegetation to another type of ecosystem, namely an increased frequency and intensity of human disturbance, a higher propagule pressure and a reduction in the population size of many native species, hampering ecosystems' ability to pose biotic resistance against the newcomers.

However, within converted ecosystems, intensity effects are less consistent, with non-native species numbers even decreasing with intensity in some of the LU-types. Despite being counterintuitive at first glance, this finding could be explained by the fact that the intensification of already transformed ecosystems can reduce resource availability and habitat suitability for both native and non-native species, e.g. increased applications of pesticides. Consequently, the total numbers of species, including non-native species, can decline with increasing LU intensity for these LU-types, although non-native incidence might still increase, as such species can be more tolerant to human disturbance than natives[13]. If this were true, one might expect a hump-shaped relationship across all combinations of LU-type and LU-intensity, with particularly low non-native species numbers in ecologically intact (e.g., primary under minimal use) and heavily degraded ecosystems (e.g., intensively used plantation and cropland), and high non-native species numbers at intermediate levels of use, similar to the intermediate-disturbance hypothesis. In fact, such a hump-shaped relationship between non-native species numbers and LU intensity is already apparent to some degree for pastures and in urban areas in the PREDICTS data. Larger datasets and more in-depth analyses will, however, be required to explore this hypothesis further.

## Caveats
Clearly, the PREDICTS dataset underlying our analyses is a biased sample of local assemblages worldwide. Large areas in North America, Africa and Eastern Asia are not well covered, and some biomes are poorly represented (Supplementary Fig. 1). This is especially true for arid and cold environments and some types of grasslands and savannas[27]. On the other hand, some temperate regions, especially in Europe are densely sampled (Fig. 1a). Moreover, the spatial coverage of different geographical areas and biomes is uneven across taxa (Supplementary Fig. 1), with large gaps especially in ants (few temperate assemblages) and spiders (few tropical assemblages). Differences in the levels of invasion among groups (Fig. 1) may be related to these spatial biases, because regional pools of non-native species also show pronounced geographical variability[45]. However, concerning the response of non-native incidence, number and proportions to LU, our

sensitivity analyses do not suggest major variation in the patterns detected among the biomes covered by the data. This does not exclude future studies with more complete and representative spatial coverage of assemblages might lead to some revision of our findings. In particular, some apparently idiosyncratic patterns in our data, such as the peak of spider incidence in pastures, may disappear. In contrast, some other patterns, which were not statistically significant in our analyses (e.g. LU effects on non-native proportions) or could not even be included in the analysis due to low sample size (e.g. mammal incidence in urban areas), might become detectable.

Given that almost three quarters of the terrestrial surface of the planet is estimated to have been subject to human land transformation already[51], primary vegetation is likely over-represented in the PREDICTS data (Supplementary Table 16). This over-representation may result in an underestimation of the average non-native contribution to local assemblages. On the other hand, assemblages from the vast cold and arid regions of the globe are also poorly represented, and these are often sparsely invaded[14]. Further, if non-native species are much rarer than native species in a study site, they will be more likely to have been missed or misidentified during sampling, as they are not expected to be present there. Sampling efforts in the original studies underlying PREDICTS may have also differed across LU-types, taxa and biomes[3]. For instance, rare non-native species may have been difficult to detect in remote primary vegetation with low accessibility, even if sampling protocols were standardised within individual studies[26]. In addition, site selection in the individual studies compiled in PREDICTS may have artificially increased differences in non-native frequency among primary vegetation and other LU-types. This is particularly likely in the case of plants, where the presence or absence of non-native species may have been used among the criteria to identify primary vegetation. Finally, regional databases of naturalised non-native species are certainly incomplete[45] and species lists of assemblages in PREDICTS are based on different sampling methods and intensities[26]. Gaps and inconsistencies in the two data sources underlying our analysis likely propagate into our results, introducing 'noise' that may mask signals in the analyses.

Our results demonstrate that non-native species have already established populations in many local assemblages worldwide. While many of these species have negligible effects on the invaded ecosystems or may functionally replace lost native ones[52], a fraction of them has negative ecological impacts[14] and entail considerable economic costs[53]. As our findings suggest, the degree to which particular LU-types and the intensity of management facilitate the local establishment, and hence eventually the regional spread, of non-native species varies and is not entirely consistent among taxonomic groups. However, across all taxonomic groups, primary vegetation is least invaded, or at least not more invaded than any other LU-type. We hence conclude that the conservation or restoration of natural and near-natural ecosystems may constrain, to some extent, the introduction and spread of biological invaders, while it simultaneously serves as a measure of (native) biodiversity conservation[46]. Conversely, if growing human populations and economies drive further land conversion and LU intensification, this will not only have negative effects on native species, but will also promote the spread of non-natives across local assemblages, especially if global trade and traffic continue to accelerate the transcontinental exchange of species[45,54]. These results additionally underline the need to restrict further natural land conversion, and to support ambitious conservation and restoration goals.

## Methods
### Data collection
**Local assemblages and different LU-types and LU-intensity levels.**
Data on local species assemblages were obtained from the published PREDICTS database[27]. PREDICTS is a compilation of data from 666 original studies published in 480 different sources (sampling period 1984–2013) and contains species lists of 26,114 local assemblages. Assemblages are defined as sets of species sampled by the authors of the original studies according to study-specific sampling designs, most often plots or transects of varying size (linear extents of sampling areas vary between 0.06 and 39,150 m, with median value of 60 m). Each individual study compares assemblages sampled from habitats characterised by different LU-types and/or LU-intensities[26] using a single sampling approach. Sampling effort was also equal at all sites within the majority of studies and varies but little, and for reasons other than LU (e.g., trap damage), within the remainder. When publications reported the data from multiple different sampling approaches, the data from each sampling approach has been added to the PREDICTS database as a distinct study from within the same source. All assemblages were assigned to one of eight LU-types and one of three LU-intensity levels by the PREDICTS team, based on information provided in the original studies[3] (Supplementary Table 1). The assemblage's assigned LU-type is the LU class that best describes the land use in the sampling frame or, if the maximum linear extent sampled is less than 10 m, the 100 m² centred around the sampling frame. The LU-types considered were primary vegetation (Primary); secondary vegetation (Secondary, three stages of secondary succession (young/intermediate/mature), which we have combined into one for this study); Plantation (i.e. permanent crop trees for fruit and/or wood harvest, such as coffee, oil palm and timber); Pasture (i.e. converted or natural grasslands used for livestock grazing); Cropland (i.e. planted with herbaceous crops, such as wheat) and Urban (i.e. areas with human habitation and/or buildings). The LU-intensity levels distinguished were Minimal use, Light use and Intense use (Supplementary Table 1). The PREDICTS database contains geographically nested study-site blocks (SSB), nested in study sites (SS)[26].

**Regional distributions of non-native species.** We considered five taxonomic groups for which information on non-native species naturalised at the regional scale was available in established databases: ants[55], birds[47], mammals[31], spiders[56] and vascular plants[17,57]. The latest versions of these databases document 303 ant, 361 bird, 239 mammal, 207 spider, and 15,111 vascular plant species naturalised as non-native in regions stated in the databases, respectively (details on the data sources are provided in Supplementary Table 2). Species distributions in the databases were assigned to the Biodiversity Information standards (TDWG level 4), which includes 609 terrestrial regions (186 islands or archipelagos, and 423 mainland regions, mostly countries, states and provinces of larger countries)[58].

**Combining local assemblages and regional non-native species distributions.** PREDICTS includes data on the species composition of local assemblages at different levels of taxonomic rank. While most studies contain complete species lists for larger taxonomic groups/ranks, some had a narrower focus on particular families, genera or even individual target species. To avoid bias in the estimates of non-native species incidence or number (see below), we excluded studies with a taxonomic rank below the one of the non-native species databases described above. We therefore excluded 196 studies referring to taxonomic groups lower in rank than Mammalia, Passeriformes, Araneae, Formicidae, or Magnoliopsida. To avoid losing assemblages of our five groups nested within higher ranks in PREDICTS, we also included studies referring to Animalia, Chordata, Aves, Arthropoda, Arachnida, Insecta, Hymenoptera and Tracheophyta. From the assemblages in the latter studies, we then selected the species lists of Mammalia, Aves, Formicidae, Araneae and Tracheophyta. The status of the species in the assemblages, for example whether it is breeding or not, is not consistently documented and could hence not be considered in the analysis.

Species names across all datasets were standardised to the taxonomic backbone of the Catalogue of Life, a comprehensive

database including c. 80% of all species known to science (2.4 million)[59]. Species name standardisation was conducted by using the *rcol* R package version 0.2.0[60], and included the following steps: First, species (e.g. synonyms, subspecies and variants) were standardised to accepted scientific names. Second, we manually checked spelling for unresolved cases. For cases still unresolved, we consulted experts in the specific taxonomic group. Finally, 154 species-entries in PREDICTS (30 ants, 7 birds, 21 spiders and 96 vascular plants) with invalid names that we could not assign to an accepted taxon even with the help of the experts were removed from the analyses (Supplementary Table 3).

The locations of local assemblages were assigned to TDWG level 4 regions based on the geographical coordinates published with each assemblage (and collated in PREDICTS). Based on this assignment, we identified the species as non-native to a particular region in each assemblage by matching species lists of local assemblages with the lists of species from the non-native distribution databases for the assemblage's region. Assemblages falling into a TDWG region not included in the regional non-native distribution database of the respective taxonomic group were excluded from the analysis, i.e. we only used assemblages from the PREDICTS database which are located in TDWG level 4 regions with non-native species information. For each assemblage, we then calculated the total number of species, the number of non-native species, and the non-native proportion (ratio of non-native to total number of species).

### Data analysis

As the majority of assemblages did not contain a single non-native species (see below), we analysed the data via a two-step approach, similar to what is achieved in a hurdle model. First, we modelled the probability of encountering at least one non-native species in an assemblage using the full dataset. Second, we modelled both the number of non-native species and the proportion of non-natives among all species in the subset of assemblages containing at least one non-native species. For each of these three response variables, we fitted separate initial generalised linear mixed effects models with the predictor variables (a) LU-type and LU-intensity, together with their interaction, and (b) LU-type alone. In models including LU-intensity, not all assemblages could be used as some had not been assigned to any of the intensity levels by the PREDICTS team[26,27] (intensity level "Cannot decide"). In models including LU-type only, all assemblages were considered (including the ones with intensity level "Cannot decide"), and we fitted models across all taxa and for each taxon separately. The assemblages of ants, mammals, spiders and vascular plants in urban areas, and mammals in cropland could not be included as the number of assemblages for these levels was too low. Each model was initially fitted with the full random effects for study-site blocks (SSB) nested in study sites (SS)[3]. Stepwise backward model selection was performed using the Akaike Information Criterion (AIC) and likelihood ratio tests, and in each step, we selected the simpler model if its AIC was at least 2 points smaller than the more complex ones and the *p*-value of the likelihood ratio test comparing the two models was <0.05. We started by selecting the random effect structure first (with constant full fixed effects structure), and then selected the fixed effects structure (with random effects as selected in the previous step). Finally, we used the *emmeans* package version 1.7.2[61] to compute back-transformed model estimates (i.e. absolute modelled values in response scale). For details on each response variable and its models, see below.

**Non-native incidence.** To analyse the effects of LU-type and LU-intensity on the probability of encountering at least one non-native species (non-native incidence), we created a binomial response variable (zero vs. at least one non-native species) and modelled it with logistic generalised linear mixed effects models using the glmer function from the *lme4* package version 1.1-27.1 with a binomial error distribution[62]. To aid model convergence, the LU-type and intensity combination 'Urban under Intense use', and the LU-type plantation for taxonomic group spiders had to be removed as they led to complete separation within the respective models (each assemblage in these predictor level combinations contained at least one non-native species, and we took these levels to have a non-native incidence of 1).

Prior to model selection, we tested for over- or underdispersion with the *DHARMa* package version 0.4.5[63]. The test result indicated our data met the assumptions of the fitted interaction model ($p = 0.53$) and of the fitted LU-type only models ($p = 0.54$ across all taxa and $p = 0.67$, 0.79, 0.78, 0.86 and 0.94 for the ants, birds, mammals, spiders and vascular plants; respectively). The final, minimal models, i.e. after model selection, contained the full random and fixed effect structures as described above, i.e. no model parameter was removed. We computed back-transformed pairwise contrasts (i.e. effect sizes relative to the reference level) and model estimates with Tukey correction of *p*-values with the *emmeans* package. We present the effect sizes of each predictor level combination as the odds ratios of encountering a non-native species in an assemblage under a particular land use compared to the reference level 'Primary under Minimal use' in the interaction model, and 'Primary' in the LU-type only model.

**Non-native number and proportion.** To analyse the effects of our predictor variables on the number of non-native species, we fitted generalised linear mixed effects models using the glmer function from the *lme4* package version 1.1-27.1 with a Poisson error distribution[62]. We detected neither over- nor under-dispersion of the interaction model ($p = 0.24$) nor of the LU-type only models ($p = 0.72$ across taxa and $p = 0.17$ and 0.77 for the ants and vascular plants; respectively). However, the DHARMa tests for the LU-type only model for birds and spiders indicated over- and underdispersion, respectively. We thus switched to the *glmmTMB* package version 1.1.2.3 with a compois distribution[64] for these two taxa's models ($p = 0.24$ and 0.1 for the DHARMa tests; respectively). For mammals, number of non-native species in response to LU-type was not analysed by a statistical model since the majority of assemblages (87.4%) have one non-native species. Model selection retained the full set of random and fixed effects terms for the final interaction model. The final LU-type only models contained the full random effects structure across all taxa for vascular plants, but only a random factor for sites (SS) for ants, birds and spiders, respectively.

We modelled non-native proportions with generalised linear mixed effects models using the glmmTMB function from *glmmTMB* package version 1.1.2.3 with a beta error distribution[64], which is able to correct for over- and under-dispersion which was detected with the *DHARMa* package version 0.4.5[63] for all analogous models using the glmer function. Some assemblages have a non-native proportion of 1, i.e. all species present in those assemblages were non-native. The beta distribution, however, does not cover the value 1, and we applied the so-called lemon-squeezer transformation prior to fitting the model[65]. We detected neither over- nor under-dispersion of the interaction model ($p = 0.1$) nor of any of the LU-type only models ($p = 0.14$ across taxa and $p = 0.06$, 0.38, 0.41, 0.13 and 0.98 for the ants, birds, mammals, spiders and vascular plants; respectively). The final minimal interaction model contained the LU-type and -intensity interaction, but a simplified random effects structure including only study sites (SS). The final minimal LU-type only models contained the full random effects structure for birds and vascular plants, but only a random factor for sites (SS) across all taxa, ants, mammals and spiders, respectively.

For both the final non-native count and proportion models, we present model estimates back-transformed with the *emmeans* package[61], and Tukey corrected pairwise comparisons across all factors that were used to determine statistical significance ($p < 0.05$).

## Sensitivity analysis

We tested our model results for sensitivity to (1) variation in the spatial coverage of different biomes in the PREDICTS data; (2) variation in the size of the sampling area across the PREDICTS assemblages; and (3) the fact whether the assemblages were sampled on an island or not. For the first of these three points, we combined two different approaches. First, we introduced the location of each assemblage in one of the 14 biomes distinguished by Olson et al[66] as an additional nested random effect in all models, i.e. we replaced the random-effects structure of SSB/SS with SSB/SS/biome, or, in cases where model selection had favoured a simpler random-effects structure (see "Data analysis" above), we replaced SS by SS/biome. We then compared the respective models with and without biome in the random effects using the Akaike Information Criterion (AIC) and likelihood ratio tests. Second, we performed a leave-one-out cross-validation where we re-fitted each model excluding all assemblages from one biome in turn. We subsequently calculated the means of all regression coefficients, as well as their standard errors and their 95% confidence intervals. Coefficient estimates from all models were back-transformed pairwise contrasts to the reference level 'primary vegetation under Minimal use' in full models (with fixed effect terms LU-type * LU-intensity) and the reference level 'primary vegetation' in models with LU-type as the only fixed-effects predictor. In the case of non-native species numbers and proportions, spiders could not be analysed separately because of the low sample size in several biomes.

The sensitivity of model results to differences in the size of the sampling area was evaluated by using the sampling area size as an additional fixed-effects predictor in the models, i.e. using the fixed-effects combination LU-type * LU-intensity + sampling area. We then evaluated whether the sampling area had a significant effect on non-native incidence, number or proportion in these models and whether its inclusion changed the significance of any other predictor in the models. Before fitting the models, the linear extents of sampling areas of local assemblages were log-transferred to increase the symmetry of their distribution.

Finally, we assessed whether non-native incidence, number and proportion were responding differently to LU on islands vs continents by using location on an island as an additional random factor in the models, in the same way as described above for location of assemblages in particular biomes, i.e. by replacing the random effects structure SSB/SS by SSB/SS/Island-Continent or SS by SS/Island-Continent. We identified the assemblages on an island based on a worldwide island dataset (19,392 islands > 1 km²)[67]. Fractions of assemblages from islands were (44.4%, 20.9%, 17%, 15%, 17.7% for spiders, ants, birds, mammals and vascular plants; respectively). Second, we ran the models for non-native incidence, non-native number and proportions by excluding the assemblages from islands. Subsequently, we calculated the means of all regression coefficients and their standard errors for the two models, the one excluding island assemblages and the one including all assemblages (confidence interval was not calculated here since there were only two models).

All analyses were conducted in R version 4.1.1 (R Core Team 2021) and figures were produced using the package *ggplot2* version 3.3.5[68].

### Reporting summary

Further information on research design is available in the Nature Portfolio Reporting Summary linked to this article.

## Data availability

Data are available at https://github.com/liudyuk/data-and-code-for-alien-invasion-in-local-assemblage.git. The data.csv file is for the statistical analyses and the files in the folder of "Data used in the analysis" are data for ants, birds, mammals, spiders and vascular plants used to produce our results, i.e. the lists of species identified as alien in the PREDICTS assemblages.

## Code availability

Codes for the analyses and figure are available for this paper at "https://github.com/liudyuk/data-and-code-for-alien-invasion-in-local-assemblage.git".

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

## Acknowledgements

We appreciate all the people who conduct field measurements and contribute to the PREDICTS dataset. We also greatly thank Dr. Tim Newbold for help and explain the detailed information on the PREDICTS database. M.v.K. and M.W. thank the German Research

Foundation DFG for funding (M.v.K.: grant 264740629; M.W. via iDiv: FZT 118, 202548816). H.S. received funding through the 2017–2018 Belmont Forum and BiodivERsA joint call for research proposals, under the BiodivScen ERA-Net COFUND programme, and with the funding organisation BMBF (grant number 16LC1807A). C.C. was supported by Portuguese National Funds through Fundação para a Ciência e a Tecnologia (CEECIND/02037/2017). B.L. and F.E. was supported by the Austrian Science Foundation FWF (grant I2086-B29). PP and JP were supported by EXPRO grant no. 19-28807X (Czech Science Foundation) and long-term research development project RVO 67985939 (Czech Academy of Sciences). D.L. is financially supported in whole by the FWF Austrian Science Foundation (Lise Meitner Programme M2714-B29).

## Author contributions

S.D. designed the research and D.L. conceived the ideas, coordinated data collation, and designed and led the analyses and writing with major revision from S.D., P.S., F.E., B.L., D.M., H.S. and further inputs from all other authors. The GloNAF core team (W.D., F.E., M.v.K., P.P., J.P., M.W., H.K. and P.W.) and B.L. contributed the database for vascular plants, E.P.E. and B.G. for ants, T.M.B. and E.E.D. for birds, D.B., C.R. and P.C. for mammals, and W.N. for spiders.

## Competing interests

The authors declare no competing interests.

## Additional information

[1]Department of Botany and Biodiversity Research, University of Vienna, Rennweg 14, 1030 Vienna, Austria. [2]Department of Arctic Biology, UNIS–The University Centre in Svalbard, 9171 Longyearbyen, Norway. [3]Research Department of Genetics, Evolution and Environment, University College London, London, UK. [4]Institute of Zoology, Zoological Society of London, London, UK. [5]Invasion Science and Wildlife Ecology Lab, School of Biological Sciences, The University of Adelaide, Adelaide, SA 5005, Australia. [6]Global Mammal Assessment programme, Dipartimento di Biologia e Biotecnologie "Charles Darwin", Sapienza Università di Roma, Rome, Italy. [7]Centro de Estudos Geográficos, Instituto de Geografia e Ordenamento do Território da Universidade de Lisboa, Lisboa, Portugal. [8]Laboratório Associado TERRA, Tapada da Ajuda, 1349-017 Lisboa, Portugal. [9]Department of Biosciences, Durham University, South Road, Durham DH1 3LE, UK. [10]UK Centre for Ecology and Hydrology, Wallingford, UK. [11]Centre for Biodiversity and Environment Research, Department of Genetics, Evolution, and Environment, University College London, London, UK. [12]Insect Biodiversity and Biogeography Laboratory, School of Biological Sciences, The University of Hong Kong, Pok Fu Lam Rd, Lung Fu Shan, Hong Kong SAR, China. [13]Biodiversity and Biocomplexity Unit, Okinawa Institute of Science and Technology Graduate University, Onna, Okinawa 904-0495, Japan. [14]Radcliffe Institute for Advanced Study, Harvard University, Cambridge, MA 02138, USA. [15]Biodiversity, Macroecology & Biogeography, University of Göttingen, Büsgenweg 1, D-37077 Göttingen, Germany. [16]Centre of Biodiversity and Sustainable Land Use (CBL), University of Göttingen, Büsgenweg 1, D-37077 Göttingen, Germany. [17]Czech Academy of Sciences, Institute of Botany, Department of Invasion Ecology, CZ-252 43 Průhonice, Czech Republic. [18]Department of Ecology, Faculty of Science, Charles University, Viničná 7, CZ-128 44 Prague, Czech Republic. [19]Ecology, Department of Biology, University of Konstanz, Universitätsstrasse 10, D-78457 Konstanz, Germany. [20]Zhejiang Provincial Key Laboratory of Plant Evolutionary Ecology and Conservation, Taizhou University, Taizhou 318000, China. [21]Institute of Ecology and Evolution, University of Bern, Baltzerstrasse 6, CH-3012 Bern, Switzerland. [22]Senckenberg Biodiversity and Climate Research Centre, Senckenberganlage 25, 60325 Frankfurt, Germany. [23]Campus-Institut Data Science, University of Göttingen, Goldschmidtstraße 1, D-37077 Göttingen, Germany. [24]German Centre for Integrative Biodiversity Research (iDiv) Halle-Jena-Leipzig, Leipzig, Germany. [25]Department of Life Sciences, Natural History Museum, London SW7 5BD, UK. [26]Department of Life Sciences, Imperial College London, Ascot SL5 7PY, UK. [27]Present address: National Research Council of Italy - Institute for Bioeconomy (CNR-IBE), Via dei Taurini 19, Rome, Italy. ✉e-mail: daijun.liu@univie.ac.at

