## [Peer Review File · Nature Communications]

REVIEWER COMMENTS

Reviewer #1 (Remarks to the Author):

The authors analyzed the impact of LU types and intensity on alien species incidence and number in local assemblages worldwide for different taxonomic groups. I think the paper is nice, I like the results but I am a little bit confused about the objective of the study and the emphasize given by the authors on the % of assemblages in which alien species occurred.

My main concern is related to the objective of the study.

The authors acknowledge that most of the study exploring how LU affects alien species number in local communities are bias towards Europe and plants (L1107-120), and thus “whether these patterns hold true beyond these regions and taxa remains largely unknown”. In fact, the authors did a great job at analyzing the results but they did not explore specifically the spatial patterns. This may constitute a fourth objective (d) L130. I believe the authors need to conduct a spatial comparison of their results and better emphasize the (absence of) differences compared to temperate regions within the results and discussion. The results and discussion associated with “one fifth of local assemblages contained alien species” L249-269 are a little bit out of scope here as it does not inform about the LU-types relationships with alien. L270-281 could be discussed through the caveat.

In addition, most of the results showed differences between primary vs. the other LU-types, which is also the main message of the paper L79-81 and L244-247 but the figures presented all LU-types and it is not easy to identify the differences from the figures. I understand that the authors want to make the distinction between the LU-types but at the end they do not rely much of this information to present their findings. I would suggest to also present the results from primary vs. other LU-types across the different metrics of alien occurrences to strengthen the message of the paper.

I also think the results need to be balanced towards the taxonomic bias. L166 the authors state that the LU types only model for all taxonomic groups demonstrated that all LU types had significantly higher odds ratios of alien incidence than primary vegetation. But the differences among taxa are strong and this trend seems to be mostly driven by plants and birds (for which most data are available). This is acknowledging within the discussion, but the results section could be more balanced.

The results of Fig 4a and c are not discussed in the text and only the results related to the taxonomic variations are mentioned. I also suggest to reorganize the results by combining Fig 4 a and c with

results from Fig 3. Those are the same results but the focus is on LU-intensity and then the authors can discuss about the taxonomic differences using Fig 4bd.

I need the abstract needs to be re-written regarding the main results part. When first reading the abstract I was quite disappointed by the content. For instance, I don't think that result from L77-78 should be emphasized in the abstract (i.e., % of sampled assemblages with at least one alien species). Moreover, "the degree to which particular land use types ... varies across taxonomic groups" is very vague statement and does not give actual content about the role of land use, only L80-81 does. I think the authors showed a lot of results in the main ms and the abstract could better reflect the main findings. In addition, the abstract justifies the need to conduct such a study because "knowledge is currently driven towards vascular plants in temperate regions", but the abstract does not give much information if the findings differ from temperate regions or plants.

My second main concern is related to the database itself. The authors compared the LU-types of very different organisms (birds, plants, mammals, ants, spiders). I was wondering what is the resolution of the LU-type information. I did not find this information in the method section. Is the LU-type assigned based on the assemblage coordinate, is it a buffer around the assemblage? Do the author know if the LU-type around the assemblage may impact the alien occurrences?

L430: I am not sure that I understand how the authors combined PREDICTS database and regional alien species distributions. It seems that the alien occurrences are not directly extracted from PREDICTS but from an overlap between PREDICTS and another database of spatial information. Those are not "true assemblage", this is fine, but it should be mention in the discussion as one of the main caveat, it might explain why most of the assemblages only have one alien species.

Related to that I don't know what is the policy of the journal regarding data availability, but the dataset combining PREDICTS and alien occurrences should be made available to ensure the reproductability of the results.

Minor comment:

L94 : I think a word is missing between individual driver and the interaction.

L242-244 The statement is strong given the findings of the study this is not true for the majority of taxa

Reviewer #2 (Remarks to the Author):

The manuscript “The impact of land use on alien species incidence and number in local assemblages worldwide” investigates the degree to which NIS species are incorporated in local “assemblages” within 5 taxa, and determine if there is a relationship with landuse type and intensity.

Below, I provide comments as encountered in a couple reading of this manuscript.

Abstract: The term alien species is immediately off-putting. The terms alien and exotic species are not scientifically descriptive, and are loaded terms that should not be used. Non-indigenous species (NIS) or non-native species are much preferred terms, and these may be introduced to an area or may spread to an area (invasive).

Line 71: Refers to spread, but spread not assessed at all. Really just occurrence in a particular place, may have spread may have been directly introduced.

72, 73: Plenty of studies on birds and mammals.

74: And throughout – local assemblages are not defined as to spatial and temporal scale of interest. I’ve looked through the PREDICTS database, and this is not clear there either, to me. Assemblages are often just what was sampled opportunistically, and rarely corresponds with a functioning community. Also, recording a species as present says nothing about its establishment at a particular location. So, I have considerable trouble with this data, and the lack of description of it.

77: 21% with NIS - - I’m suspicious of this, a very low number. But can’t assess because of the comment above.

80: What is a primary ecosystem?

90: Versus exploitation of unnatural populations? Also, ecosystem modification began long before the Anthropocene.

96: LU Change - - really habitat loss more specifically, LU change excludes LC change, though they overlap.

99: “used ones”? All ecosystems are used by humans in one way or another.

100: Defines alien as introduced. So really just introduced species (in which case use this term), excluding spp. that are non-native but invaded (spread into site)?

102: I.e., are r-selected, human commensals, or early colonist spp?

105-106: Lots of evidence besides "long-hypothesized". No review attempt.

107: "Alien spp numbers" - - I believe authors do not mean #s per se (abundance of each spp), but rather species richness.

113-120: NIS introduction, establishment, and spread, may (and do) have different drivers.

121-122: Limited land use types. Intensity and LU are likely NOT independent.

124: How linked?

125: What is an assemblage? Spatial extent, temporal duration. Was scale same for each assemblage? Same sampling intensity? Established (breeding) species only? Why these LU types, these particular assemblages, selected, and how?

139 "assemblage samples" Were these then sampled, or just referring to the assemblages.

139: Lists - - what type of list. Species lists from local site often are pretty poor. Breeding spp only? Everything observed? Transients? Wintering species (birds)? Same scale?

145: Having the majority of assemblages have no non-native species makes me suspect these are very limited "lists".

148-149: Looks like oceanic islands were included - - these are very special cases when it comes to non-native species.

- I also can't think of any North American assemblage (except perhaps in the far North) that don't have at least one non-native ant or bird.

164: Croplands and urban areas of course may have nearly no native vegetation, and croplands are fully coerced.

168-170: Not clear to me how one would compare intensity across such disparate LU types. Is low intensity agriculture equivalent to high intensity "primary vegetation"?

222: Birds in croplands - I suppose it depends which, and what one considers to be present - but few birds would actually be resident in intensive croplands, utilizing them yes, but not resident per se. Again, begs the question of what is an assemblage?

238: Discussion covers 30% of this paper. Maybe ok...

240-241: Not true, at least at better resolution, habitats, communities, ecosystems, countries.

254: Seems a stretch, and only true if the assemblages used were the same size, had the same data collection duration and intensity and METHODS.

258: 24 spp in a tiny "assemblage". Seems like the scale of analysis does not match the scale of data. How many avg for birds and other taxa?

272: vascular plants most sensitive - - no citation for this.

280: mammals have been introduced a fair bit, but rarely are un-noticed as a “stow away”, especially compared to ants, e.g.

299, 300: What does this mean?

399: Land “consumption” – not mentioned before, what does this mean?

406; richness and composition – what does composition mean? ID of the spp?

411: Some of these are landcover types, not land use types.

423: Low numbers.

425: Mapped? Meaning?

433: So lists vary as to resolution even within taxa?

454: Lots of potential error here.

462: Point to the narrowness/incompleteness of these lists.

480: Seems to be mixing information theoretic approaches with frequentist statistics. And, AIC already penalizes for more variables.

Reviewer #3 (Remarks to the Author):

This is a very timely study focusing on the global pattern of the interaction between land-use change and biological invasions. The authors used a large dataset across five taxonomic groups and analyzed the levels of invasion and the relationship between the land use intensity and the alien species distributions in local assemblages worldwide. Their findings provide new global and quantitative evidences on the importance of primary vegetation conservation that may help constrain alien species establishment. The authors also discussed their results very carefully and did not overstate their findings considering the sampling bias issues in the database and similar macroecology studies. I enjoy reading the manuscript and find it is a great work with a general topic suitable for the broad readership of Nature Communications. I have some specific suggestions that hopefully can help with the improvement of the work. I am also signing this review, and I am available for further review of revisions if necessary.

Introduction section

Line 72-73, Line 102 : except for the studies of vascular plants in temperate regions, but also see large-scale spatial analysis on the relationship between other animal invaders such as the bullfrog distribution and land-use change (Ficetola et al. 2010), and local-scale study on the accelerating range expansion of the bullfrog in human disturbed habitats (Wang et al. 2022).

Line 76, Line 127: I notice that the authors have provided detailed explanations on how you define the incidence in the method section. However, it is a little difficult for readers especially for those general readers to understand when it appears in the manuscript for the first time. Which invasion stage does it indicate here? Does it mean the establishment of alien species or just the presence of the alien species? Please clarify.

Line 77, 127: The authors need to define the intensity of land use more clearly because the variables reflecting the change of land-use is very diverse, which have different ecological implications (e.g., Semenchuk et al. 2022). For example, how about the potentially different effects of land conversion and land-use intensity on alien species invasions? I think it at least warrants some discussions.

Line 133-135: It is interesting to see the authors expectation that plants may be more responsive to land use. But it is better to provide more theoretical basis for this expectation here.

Result section

Line 144-147: In addition to the proportion of local assemblages containing only one alien species, it will be better to describe the distributions with more than one alien species among habitat types across taxa despite some of this information have been included in Fig. 1.

Discussion section

Line 307-327: Regarding the variations in invasion patterns among habitats with different ecosystems, please provide more potential mechanisms behind the pattern. The authors have tried to explain this from the resource availability and introduction pathways. Moreover, does the low degree of invasion in primary ecosystems just reflect the low human activity? Will there be any other alternative mechanisms or explanations? For example, are there more native species in human less-disturbed habitats and thus may have higher biotic resistance to biological invaders despite this hypothesis remain controversial (Fridley et al. 2007)? In addition, whether the number of historical already established alien species will facilitate more alien species invasions based on the invasion meltdown hypothesis (Simberloff & Von Holle 1999, Redding et al. 2019)?

Line 364-383: It is good that the authors have addressed the sampling bias issue. However, in addition to this issue behind the observed taxonomic mismatches, there might also be different sampling efforts among land-use types. In theory, the invasive alien species may be more likely to be recorded in habitats with higher human activities as we can conduct surveys in these habitats more easily than those natural primary vegetation habitats in remote areas.

References in this review

Ficetola, G. F. et al. Knowing the past to predict the future: land-use change and the distribution of invasive bullfrogs. *Global Change Biol* 16, 528-537, (2010).

Fridley, J. D. et al. The invasion paradox: Reconciling pattern and process in species invasions. *Ecology* 88, 3-17 (2007).

Redding, D. W. et al. Location-level processes drive the establishment of alien bird populations worldwide. *Nature* 571, 103-106, (2019).

Semenchuk, P. et al. Relative effects of land conversion and land-use intensity on terrestrial vertebrate diversity. *Nature Communications* 13, (2022).

Simberloff, D. & Von Holle, B. Positive interactions of nonindigenous species: invasional meltdown? *Biol Invasions* 1, 21-32 (1999).

Wang, X. et al. Anthropogenic habitat loss accelerates the range expansion of a global invader. *Divers Distrib*, doi: <https://doi.org/10.1111/ddi.13359> (2022).

Response to reviewers

REVIEWER COMMENTS

Reviewer #1 (Remarks to the Author):

The authors analyzed the impact of LU types and intensity on alien species incidence and number in local assemblages worldwide for different taxonomic groups. I think the paper is nice, I like the results but I am a little bit confused about the objective of the study and the emphasize given by the authors on the % of assemblages in which alien species occurred.

We thank the reviewer for the positive assessment of our paper. We agree that this particular result on the % of assemblages has received too much space in the manuscript, especially in the Discussion section. Therefore, we have shortened this part (while, at the same time, adding some information in the response to comments of Referee #2). However, we would like to emphasize that this is the first global overview on alien incidence in local assemblages. We hence think that the incidence of aliens we found is a descriptive but interesting result which deserves reporting and some discussion in the manuscript.

My main concern is related to the objective of the study.

The authors acknowledge that most of the study exploring how LU affects alien species number in local communities are bias towards Europe and plants (L1107-120), and thus “whether these patterns hold true beyond these regions and taxa remains largely unknown”. In fact, the authors did a great job at analyzing the results but they did not explore specifically the spatial patterns. This may constitute a fourth objective (d) L130. I believe the authors need to conduct a spatial comparison of their results and better emphasize the (absence of) differences compared to temperate regions within the results and discussion. The results and discussion associated with “one fifth of local assemblages contained alien species” L249-269 are a little bit out of scope here as it does not inform about the LU-types relationships with alien. L270-281 could be discussed through the caveat.

While we think that resolving the analysis to a finer spatial scope, e.g. continents or biomes, is interesting we are reluctant to do so, primarily for data-limitation reasons. Except for vascular plants and birds, the number of assemblages available for analysis is often rather limited, especially when analysing the numbers and proportions of alien species (cf. Table S4). Even without splitting into regions, we could not analyse several combinations of taxonomic groups and LU-types due to data shortage. Crossing LU types and geographical regions in the analysis would greatly exacerbate this problem, i.e. for many factor-combinations the data underlying the analysis would be too sparse, or, if the analysis is technically possible, the results would be of questionable reliability. We hence refrain from using any kind of geographical structuring in the manuscript.

However, out of curiosity regarding what might emerge with a more complete dataset, we tried to include a variable “Biome” with three classes (‘Tropical’, ‘Temperate’, “Arctic”) as an additional nested random effect into the model for alien incidence. We found that this additional grouping did not improve the models for any of the taxonomic groups (all AIC-differences ≤ 2) and the fixed-effects results were unchanged. So, at least with the data available, the results do not appear to differ significantly between biomes.

As argued in our response to the first comment, we think that a description of how many of the assemblages actually contain alien species is likely to interest many readers. It is, as far as

we know, the first paper that analyses alien incidence in local communities worldwide, hence such numbers are certainly informative on their own. For this reason, we keep the parts referring to this result in the manuscript, although we shortened the corresponding paragraphs in the Discussion and removed the respective numbers from the Abstract and Conclusions sections.

In addition, most of the results showed differences between primary vs. the other LU-types, which is also the main message of the paper L79-81 and L244-247 but the figures presented all LU-types and it is not easy to identify the differences from the figures. I understand that the authors want to make the distinction between the LU-types but at the end they do not rely much of this information to present their findings. I would suggest to also present the results from primary vs. other LU-types across the different metrics of alien occurrences to strengthen the message of the paper.

We agree that the difference from LU is the main result we report. However, this was not a priori clear when we started the analysis, especially for groups other than plants. We hence think it is useful to have the comparisons among the other groups included. While all LU-types are represented in Figures 3-4, 'Primary', or 'Primary under minimal use' is always the reference in the Figures. The (statistical) difference to the other types or type/intensity combinations is indicated by the letters and thus visible for all metrics and groups. We think this is a useful way of presenting the results because the reader can simultaneously see whether there are significant differences from primary, whether there are differences among the other types or type/intensity combinations, and also how large these differences are in terms of the metrics (odds ratio, numbers, proportions). Put it differently, we think and hope that the information the referee rightly argues for is already present in the figures as they are. We also emphasize that, in addition to Fig. 2-4 there are also Figs. S2-S4 which provide an additional view on the results in contrast to primary vegetation.

I also think the results need to be balanced towards the taxonomic bias. L166 the authors state that the LU types only model for all taxonomic groups demonstrated that all LU types had significantly higher odds ratios of alien incidence than primary vegetation. But the differences among taxa are strong and this trend seems to be mostly driven by plants and birds (for which most data are available). This is acknowledging within the discussion, but the results section could be more balanced.

We fully agree on this point and have added the following sentences (Line 173-180):
'However, analysing the data separately for each taxonomic group demonstrates taxon-specific differences in the responses to LU-type (Fig. 2b-g), with the contrast between primary vegetation and all other types being consistently significant only in case of vascular plants. For all other taxa, there was always at least one LU-type that did not differ significantly from primary vegetation in the odds of alien species incidence. The identity of these types was different across groups. Only plantations had consistently higher alien incidence in all groups where this LU-type could be included in the model (all but spiders).'

The results of Fig 4a and c are not discussed in the text and only the results related to the taxonomic variations are mentioned. I also suggest to reorganize the results by combining Fig 4 a and c with results from Fig 3. Those are the same results but the focus is on LU-intensity and then the authors can discuss about the taxonomic differences using Fig 4bd.

We also agree on this point and have largely rewritten the former part ‘Invasion status’ of the Results following the referee’s recommendations. We also cite Fig. 4a and 4c now.

I need the abstract needs to be re-written regarding the main results part. 1) When first reading the abstract I was quite disappointed by the content. For instance, I don’t think that result from L77-78 should be emphasized in the abstract (i.e., % of sampled assemblages with at least one alien species). 2) Moreover, “the degree to which particular land use types ... varies across taxonomic groups” is very vague statement and does not give actual content about the role of land use, only L80-81 does. I think the authors showed a lot of results in the main ms and the abstract could better reflect the main findings. 3) In addition, the abstract justifies the need to conduct such a study because “knowledge is currently driven towards vascular plants in temperate regions”, but the abstract does not give much information if the findings differ from temperate regions or plants.

Again, we agree and have rewritten the Abstract completely.

My second main concern is related to the database itself. The authors compared the LU-types of very different organisms (birds, plants, mammals, ants, spiders). I was wondering what is the resolution of the LU-type information. I did not find this information in the method section. 2) Is the LU-type assigned based on the assemblage coordinate, is it a buffer around the assemblage? 3) Do the author know if the LU-type around the assemblage may impact the alien occurrences?

We have added a number of additional details describing the PREDICTS dataset.

We are not sure whether we understand the term ‘resolution’ correctly, but we think it refers to the different types and intensity levels differentiated in the PREDICTS dataset. These are, in principle, the ones we are using in this manuscript, with the exception that we have pooled the different kinds of secondary vegetation into one class. Please, see the Methods chapter for more details on this.

The categories were assigned to the assemblages by the PREDICTS-team on the basis of the description that individual study authors give about the usage of the sites where these assemblages have been sampled. They are hence directly referring to the assemblages, not to some buffer or surrounding areas. Please, see Hudson et al. (2014) for further details. We have added this information to the Methods chapter.

We agree that neighbourhood effects may play a role, unfortunately we have no data to analyse this.

L430: 1) I am not sure that I understand how the authors combined PREDICTS database and regional alien species distributions. It seems that the alien occurrences are not directly extracted from PREDICTS but from an overlap between PREDICTS and another database of spatial information. 2) Those are not “true assemblage”, this is fine, but it should be mention in the discussion as one of the main caveat, it might explain why most of the assemblages only have one alien species.

The PREDICTS database contains the lists of species for each assemblage. The databases of regional alien species pools were used to identify which species from these lists are

considered as being alien in that region. We rephrased the respective part of the Methods-section to read (Line 495-497):

“Based on this assignment, we identified the species alien to a particular region in each assemblage by matching species lists of local assemblages with the lists of alien species given in the alien distribution databases for the region in which the assemblage was sampled.”

We hope this is clearer now.

Yes, the assemblages in PREDICTS are not necessarily communities in a strict sense, but rather the sets of species recorded together in ecological samples taken at local sites. The database is a compilation of studies from around the world; sites within each study were sampled in the same way, but sampling methods vary widely among different studies. We now give this definition in the main text.

Related to that I don't know what is the policy of the journal regarding data availability, but the dataset combining PREDICTS and alien occurrences should be made available to ensure the reproductibility of the results.

The PREDICTS species lists with aliens identified are now available on github (<https://github.com/liudyuk/data-and-code-for-alien-invasion-in-local-assemblage.git>). We also put there the R code we used for analysis and making figures 1-4.

Minor comment:

L94 : I think a word is missing between individual driver and the interaction.

We have added the word “or” between individual drivers and the interaction.

L242-244 The statement is strong given the findings of the study this is not true for the majority of taxa

We agree and have toned down this statement accordingly (Line 265-268).

“As expected, human usage of ecosystems tends to facilitate the encroachment of alien species into local assemblages, although this effect was not detectable for all types of usage in all taxonomic groups.”

Reviewer #2 (Remarks to the Author):

Abstract: The term alien species is immediately off-putting. The terms alien and exotic species are not scientifically descriptive, and are loaded terms that should not be used. Non-indigenous species (NIS) or non-native species are much preferred terms, and these may be introduced to an area or may spread to an area (invasive).

We thank the reviewer for his suggestion. We see the reviewer's point, but would prefer to keep the term 'alien' because that is the scientific term used in Blackburn et al. (2011) and

adopted in relevant legislation (e.g. EU Regulation 1143/2014 on Invasive Alien Species, https://ec.europa.eu/environment/nature/invasivealien/index_en.htm). Thus, the term ‘alien species’ is well established. But if the editor insists on us using the term non-indigenous or non-native, we would also be willing to replace it.

Line 71: Refers to spread, but spread not assessed at all. Really just occurrence in a particular place, may have spread may have been directly introduced.

We agree, and replaced the term by “occurrence”.

72, 73: Plenty of studies on birds and mammals.

The Abstract has been completely re-written in response to suggestions of Referee #1, and this comment is hence no longer relevant.

74: And throughout – local assemblages are not defined as to spatial and temporal scale of interest. I’ve looked through the PREDICTS database, and this is not clear there either, to me. Assemblages are often just what was sampled opportunistically, and rarely corresponds with a functioning community. Also, recording a species as present says nothing about its establishment at a particular location. So, I have considerable trouble with this data, and the lack of description of it.

We agree that the description lacked clarity. The PREDICTS database is a compilation of hundreds of primary studies that did not sample data in a consistent and standardized way. As a consequence, the degree to which these species lists represent ‘functioning communities’ will certainly vary across studies – it is thus appropriate to talk about assemblages, not communities, and we now avoid the latter term in the manuscript. The methodological variation across studies is one reason for using a random effect of study in all statistical models. The variation in spatial grain of the assemblages is now defined in the Materials and Methods section, based on information in Hudson et al. (2014).

We also agree that recording of a species does not tell us anything about ‘establishment’ in an assemblage and we now avoid this term when we talk about the PREDICTS data. We emphasize, however, that we define alien species in an assemblage based on lists of species naturalized in a given region – hence at least at the regional level all species included in the analysis are considered established. We now make this clearer to the reader by pointing out the fact that we are talking about regionally naturalized species here and at several more places throughout the manuscript.

77: 21% with NIS - - I’m suspicious of this, a very low number. But can’t assess because of the comment above.

We agree, this appears low, and we were surprised ourselves. However, we discuss possible reasons, including sampling bias towards primary vegetation, in the Discussion section.

80: What is a primary ecosystem?

We have changed this into “primary vegetation”, which is the term used in the PREDICTS database. It refers to natural habitats, either completely untouched by human actions or by extreme natural events, where vegetation has never been completely destroyed. The definition of the land-use types and intensity levels is given in Table S1, following Hudson et al. (2014).

90: Versus exploitation of unnatural populations? Also, ecosystem modification began long before the Anthropocene.

It was meant as a contrast to livestock. We have replaced it with “*direct exploitation of organisms*”. It is true that ecosystem modification started much earlier, but we are here talking about the recent acceleration of threats to biodiversity.

96: LU Change - - really habitat loss more specifically, LU change excludes LC change, though they overlap.

We now give a more extensive and hopefully clearer formulation: “*Human LU – mainly land conversion and subsequent management for crop cultivation and livestock raising ...*”

99: “used ones”? All ecosystems are used by humans in one way or another.

Changed to “converted”.

100: Defines alien as introduced. So really just introduced species (in which case use this term), excluding spp. that are non-native but invaded (spread into site)?

At this point, we define alien species in a broad sense, as all those that have been introduced. This is in line with the unified invasion terminology framework proposed by Blackburn et al. (2011). In the analyses, we focus on the subset of aliens considered naturalized (not necessarily invasive) in a region. We make this clear at the end of the Introduction, in the Materials and Methods section, and by using the term “naturalized” repeatedly throughout the text.

102: I.e., are r-selected, human commensals, or early colonist spp?

Indeed, there are many r-selected species or ruderal strategists among them – we shortly return to this point in the Discussion section. However, there is a huge number of hypotheses to explain this bias towards disturbed ecosystems that we, for the sake of brevity, did not want to review in the Introduction.

105-106: Lots of evidence besides “long-hypothesized”. No review attempt.

We agree that there is a lot of evidence in many case studies. Apart from now citing some relevant papers (e.g. Ficetola et al., 2010; Wang et al., 2022, Redding et al., 2019), we have changed the formulation to “... *but a global-scale analysis of empirical data on how LU change and invasions interact at the scale of local assemblages has not yet been attempted.*”

107: “Alien spp numbers” - - I believe authors do not mean #s per se (abundance of each spp), but rather species richness.

It refers to “*Alien species richness*”.

113-120: NIS introduction, establishment, and spread, may (and do) have different drivers.

We fully agree. However, at this point we argue about the probability of finding alien species in different habitat types in general, independent of the invasion stage. So, we prefer not to go into detail here. Moreover, this differentiation refers rather to a larger scale, such as the regional one, rather than to a local scale. At the local scale, these stages are less clearly distinguishable.

121-122: Limited land use types. Intensity and LU are likely NOT independent.

We agree that the differentiation should ideally be more detailed. However, this is the classification system to which the PREDICTS team has assigned all the different kinds of information provided by the authors of the original studies. We also agree that there is likely dependence of type and intensity of usage, in the sense that the same intensity level might imply a different magnitude of pressure on species in different types of usage. For this reason, we fitted models with an interaction between type and intensity (see Fig. 2 and 3).

124: How linked?

We clarified that by writing (Line 130-132): “*We combine the different data sources by identifying species in the PREDICTS assemblages as alien if they are listed among those species naturalized in the respective region in the regional distribution databases.*”

125: 1) What is an assemblage? 2) Spatial extent, temporal duration. 3) Was scale same for each assemblage? Same sampling intensity? 4) Established (breeding) species only? 5) Why these LU types, these particular assemblages, selected, and how?

We have now expanded the description of the assemblage data in the Materials and Methods section (Line 433-445):

“Data on local species assemblages were obtained from the published PREDICTS database²⁷. PREDICTS is a compilation of data from 666 original studies published in 480 different sources (sampling period 1984-2013) and contains species lists of 26,114 local assemblages. Assemblages are defined as sets of species sampled by the authors of the original studies according to study-specific sampling designs, most often plots or transects of varying size (linear extents of sampling areas vary between 0.06 and 39,150 m, with median value of 60 m). Each individual study compares assemblages sampled from habitats characterized by different LU-types and/or LU-intensities²⁶ using a single sampling approach. Sampling effort was also equal at all sites within the majority of studies and varies but little, and for reasons other than land use (e.g., trap damage), within the remainder. When publications reported the data from multiple different sampling approaches, the data from each sampling approach has been added to the PREDICTS database as a distinct study from within the same source.”

We then refer to Hudson et al. (2014, 2017), where more details about the original sources and the assemblage data compiled in the PREDICTS database are given. Whether the species recorded are established (breeding) is not documented, we hence avoid talking about

‘established species’ when we talk about the PREDICTS data. A full account of the criteria for compiling assemblage data in PREDICTS is also given in Hudson et al. (2014).

139 “assemblage samples” Were these then sampled, or just referring to the assemblages.

It was referring to the assemblages, and we have therefore deleted “samples”.

139: Lists - - what type of list. Species lists from local site often are pretty poor. Breeding spp only? Everything observed? Transients? Wintering species (birds)? Same scale?

We selected those assemblages from PREDICTS that were indicated as providing complete lists of all species sampled (= all observations) of the taxonomic groups in question – see Materials and Methods / Combining local assemblages and regional alien species distributions. The status of the species in the community (breeding or not, transient) is not documented in PREDICTS and could hence not be considered in the analysis. We explicitly state that now in the same part of the Materials and Methods section.

145: Having the majority of assemblages have no non-native species makes me suspect these are very limited “lists”.

As stated above, we also expected more frequent occurrences of alien species. However, as discussed in the Discussion section, there are some plausible explanations for this, including the preponderance of lists from primary vegetation. Based on personal experience of some of the authors with (many) plant assemblages in Europe, a value of 21% is actually not so low, as the majority of plot samples in natural or semi-natural vegetation do not contain alien species.

148-149: Looks like oceanic islands were included - - these are very special cases when it comes to non-native species.

Yes, some assemblages from oceanic islands were included. However, as all statistical models have a random effect, the specificities of these systems should have been accounted for in the analysis.

- I also can’t think of any North American assemblage (except perhaps in the far North) that don’t have at least one non-native ant or bird.

The data on local assemblages for the five studied taxonomic groups in North America are rather scarce. In the case of plants, it is well documented in other studies that there are many assemblages without alien species in North America (see e.g. Kalusova et al. 2015). We have discussed the rather sparse representation of North American assemblages now in the Caveats section (Line 390-392).

“Clearly, the PREDICTS dataset underlying our analyses is a biased sample of local assemblages worldwide. Large areas in North America, Africa and Eastern Asia are not well covered, and some biomes are poorly represented (Fig. S1).”

164: Croplands and urban areas of course may have nearly no native vegetation, and croplands are fully coerced.

Yes, alien plants can accumulate in these LU-types. Accordingly, the proportions of alien species tend to be high there (see Fig. 4d). We note that these figures may still be conservative as some of the aliens in urban areas may not have been classified as naturalized in the respective region and hence were not included in the analysis. We now point that out now in the Discussion section.

168-170: Not clear to me how one would compare intensity across such disparate LU types. Is low intensity agriculture equivalent to high intensity “primary vegetation”?

We agree with the reviewer’s comment. LU-intensity levels are certainly hard to compare directly across LU-types. For this reason, we present the model that accounts for the interaction between type and intensity of use in the main text, and only show the results of the additive model in the Supplement for completeness. To emphasize this, we now expanded to (Line 181-187):

“When using LU-intensity as the only predictor in a model across all taxonomic groups, intensity levels ‘Minimal’ and ‘Light’ did not differ in the likelihood of alien incidence, but assemblages under intense use had higher odds of harbouring an alien species (Fig. S2b). In the full model with both LU-type, LU-intensity and their interaction as predictors, the effect of intensity was not consistent across the LU-types (Fig. 2a). While alien incidence increased with intensity of usage in primary vegetation, the relationship was variable in the other LU-types.”

222: 1) Birds in croplands – I suppose it depends which, and what one considers to be present – but few birds would actually be resident in intensive croplands, utilizing them yes, but not resident per se. 2) Again, begs the question of what is an assemblage?

We agree with this comment. For the question regarding what an assemblage is, please see our responses to some of the previous comments and the expanded description in the Materials and Methods section.

238: Discussion covers 30% of this paper. Maybe ok...

If the editors recommend to shorten it, we are willing to do so.

240-241: Not true, at least at better resolution, habitats, communities, ecosystems, countries.

Sorry, but we are not aware of a comparable local-grain study with global-extent. Therefore, we would be grateful if the reviewer could provide a reference.

254: Seems a stretch, and only true if the assemblages used were the same size, had the same data collection duration and intensity and METHODS.

We agree that this calculation would imply a standardized sampling methodology to be quantitatively valid. However, as explained at the end of the paragraph, this simplified calculation only serves for putting the numbers into context, and for that we still think they are valid because they ‘reverse the hierarchy’ among taxa to a certain degree. To additionally clarify the methodological caveats here, we have expanded the last sentence of the paragraph to (Line 287-292):

“Of course, sampling of species from the regional species pool into local assemblages is not a purely random process, and the PREDICTS data were not collected following a standardized sampling design, but this simple null model nevertheless puts the plain numbers in context, and suggests a taxon-specific mismatch between the accumulation of alien species in regional species pools^{28,29}, and the documented colonization of local assemblages by these species.”

258: 24spp in a tiny “assemblage”. Seems like the scale of analysis does not match the scale of data. How many avg for birds and other taxa?

A mean of 24 plant species appears quite reasonable for typical plant community data sampled at sites between 25 and 100 m². We have added the mean species number of local assemblages for the other taxa.

272: vascular plants most sensitive - - no citation for this.

There is no citation because it is one of the main findings in our study.

280: mammals have been introduced a fair bit, but rarely are un-noticed as a “stow away”, especially compared to ants, e.g.

We agree and have changed the sentence appropriately (Line 301-304).

“Hence, a subset of particularly successful species might easily and rapidly spread across many local assemblages. In addition, alien mammals are peculiar in that they have been frequently introduced intentionally³³- which may have also fostered their spread.”

299, 300: What does this mean?

Sorry, this was a remnant of internal revisions. The sentence has been removed.

399: Land “consumption” – not mentioned before, what does this mean?

We have replaced it with the term “*conversion*”.

406; richness and composition – what does composition mean? ID of the spp?

It means species lists, and we changed this accordingly. For the question about the completeness of these lists, see also the paragraph on ‘*Combining local assemblages and regional alien species distributions*’ in the Materials and Methods section.

411: Some of these are landcover types, not land use types.

We agree that the distinction is not particularly clear here. However, we prefer to follow the terminology established in other papers presenting and analysing PREDICTS data such as Hudson et al., 2014 and Newbold et al., 2015.

423: Low numbers.

The numbers for these five taxa are not large, but many of them are present in many different regions.

425: Mapped? Meaning?

We have replaced it with “assigned to”.

433: So lists vary as to resolution even within taxa?

Yes, that is correct.

454: Lots of potential error here.

We have revised this sentence as (Line 495-497): *“Based on this assignment, we identified the species alien to a particular region in each assemblage by matching species lists of local assemblages with the lists of alien species given in the alien distribution databases for the region in which the assemblage was sampled.”*

We agree that there will be errors in both data sources, leading to errors in their combination. We have hence added a sentence to the Caveats section (Line 410-414): *“Finally, regional databases of naturalized alien species are certainly incomplete²⁸ and species lists of assemblages in PREDICTS are based on different sampling methods and intensities²⁶. Gaps and inconsistencies in the two data sources underlying our analysis of course propagate into their combination and likely introduce ‘noise’ that may mask signals in the statistical models”.*

462: Point to the narrowness/incompleteness of these lists.

We have now described the data in the Methods section more extensively and also touch on the topic in the Caveats-section. We therefore feel that we do not have to insert an additional statement on the data at the beginning of the section describing the data analysis.

480: Seems to be mixing information theoretic approaches with frequentist statistics. And, AIC already penalizes for more variables.

This is correct, but we wanted to back up the selection process. In general, there is agreement between both approaches because AIC-differences < 2 are associated with p-values < 0.05 , and vice versa.

#####

Reviewer #3 (Remarks to the Author):

This is a very timely study focusing on the global pattern of the interaction between land-use change and biological invasions. The authors used a large dataset across five taxonomic groups and analyzed the levels of invasion and the relationship between the land use intensity and the alien species distributions in local assemblages worldwide. Their findings provide new global and quantitative evidences on the importance of primary vegetation conservation that may help constrain alien species establishment. The authors also discussed their results

very carefully and did not overstate their findings considering the sampling bias issues in the database and similar macroecology studies. I enjoy reading the manuscript and find it is a great work with a general topic suitable for the broad readership of Nature Communications. I have some specific suggestions that hopefully can help with the improvement of the work. I am also signing this review, and I am available for further review of revisions if necessary.

Thank you very much for your positive comments.

Introduction section

Line 72-73, Line 102 : except for the studies of vascular plants in temperate regions, but also see large-scale spatial analysis on the relationship between other animal invaders such as the bullfrog distribution and land-use change (Ficetola et al. 2010), and local-scale study on the accelerating range expansion of the bullfrog in human disturbed habitats (Wang et al. 2022).

We appreciate the reviewer's suggestion, which has now been added to (former) line 102. The Abstract has been completely re-written in response to comments of Referee #1.

Line 76, Line 127: I notice that the authors have provided detailed explanations on how you define the incidence in the method section. However, it is a little difficult for readers especially for those general readers to understand when it appears in the manuscript for the first time. Which invasion stage does it indicate here? Does it mean the establishment of alien species or just the presence of the alien species? Please clarify.

We tried to clarify this in the re-written Abstract by explicitly talking about "*naturalized alien species*".

Line 77, 127: The authors need to define the intensity of land use more clearly because the variables reflecting the change of land-use is very diverse, which have different ecological implications (e.g., Semenchuk et al. 2022). For example, how about the potentially different effects of land conversion and land-use intensity on alien species invasions? I think it at least warrants some discussions.

We agree that LU-type (implying conversion) and intensity (implying different kinds of management within the same type can have quite different effects on LU (some of the authors of this paper are also authors of the cited paper of Semenchuk et al. (2022)). For this reason, we analysed the dependence of alien species incidence/number/proportion for both different types of LU and different levels of LU-intensity. In doing that, we were bound to the differentiation of types and levels provided in the PREDICTS database (as described in the Methods section). Differences in the effects of LU-type and intensities are provided in the various figures, described in the text, and discussed in the Discussion section. We now also explicitly refer to LU-intensity effects in the Abstract, to make this distinction clear early on.

Line 133-135: It is interesting to see the authors expectation that plants may be more responsive to land use. But it is better to provide more theoretical basis for this expectation here.

We introduce this issue more broadly in the previous paragraphs of the Introduction and come back to the hypotheses on how to explain these differences in the Discussion section.

We prefer to stay with this distribution of the arguments, because in the Discussion, they serve to explain the patterns we found – which the readers do not know before they have seen the results.

Result section

Line 144-147: In addition to the proportion of local assemblages containing only one alien species, it will be better to describe the distributions with more than one alien species among habitat types across taxa despite some of this information have been included in Fig. 1.

We agree and have added the proportions of assemblages with more than one alien species for each taxonomic group to the description (Line 150-152).

“Assemblages with more than one alien species were most frequent for vascular plants (19.7%), but considerably rarer in the animal taxa analysed (ants: 6.1%, birds: 3%, mammals: 3.2%, spiders: 8.2%).”

Discussion section

Line 307-327: Regarding the variations in invasion patterns among habitats with different ecosystems, please provide more potential mechanisms behind the pattern. The authors have tried to explain this from the resource availability and introduction pathways. Moreover, does the low degree of invasion in primary ecosystems just reflect the low human activity? Will there be any other alternative mechanisms or explanations? For example, are there more native species in human less-disturbed habitats and thus may have higher biotic resistance to biological invaders despite this hypothesis remain controversial (Fridley et al. 2007)? In addition, whether the number of historical already established alien species will facilitate more alien species invasions based on the invasion meltdown hypothesis (Simberloff & Von Holle 1999, Redding et al. 2019)?

We suppose this comment refers to lines 283-306 of the original manuscript as this is the paragraph where we discuss differences in invasibility among habitats.

Following the suggestion of the referee, we have now added text that refers to several of the most common and plausible explanations for the success of alien plants in human-disturbed vs. primary ecosystems, including fluctuating resources, lower propagule pressure, and the tendency of successfully naturalized plants to be ruderal life history strategists. However, we did not go into this topic at length, because it is not central to the cross-taxon scope of our manuscript. For this reason, we also do not mention native species diversity as a potential barrier to alien invasions or the invasional meltdown hypothesis. Whether these factors generally contribute to lower (plant) invasion levels in primary vegetation is not straightforward and would require more extensive discussion (potentially also including some of the many other hypotheses put forward to explain invasions). However, the Discussion section is already quite long now (see comments of Referee #2 on this point).

Line 364-383: It is good that the authors have addressed the sampling bias issue. However, in addition to this issue behind the observed taxonomic mismatches, there might also be different sampling efforts among land-use types. In theory, the invasive alien species may be more likely to be recorded in habitats with higher human activities as we can conduct surveys in these habitats more easily than those natural primary vegetation habitats in remote areas.

While we agree with the referee in principle, this potential bias does not apply to the PREDICTS database. This is a database where original studies of local assemblages have

been compiled. In each of the studies, a comparison of the species composition of assemblages in habitats under different usage, especially primary vs. non-primary habitats, was the explicit aim, with the same sampling protocol used at all the sites within that study. Land use is therefore not conflated with sampling effort in PREDICTS.

References in this review

- Ficetola, G. F. et al. Knowing the past to predict the future: land-use change and the distribution of invasive bullfrogs. *Global Change Biol* 16, 528-537, (2010).
- Fridley, J. D. et al. The invasion paradox: Reconciling pattern and process in species invasions. *Ecology* 88, 3-17 (2007).
- Redding, D. W. et al. Location-level processes drive the establishment of alien bird populations worldwide. *Nature* 571, 103-106, (2019).
- Semenchuk, P. et al. Relative effects of land conversion and land-use intensity on terrestrial vertebrate diversity. *Nature Communications* 13, (2022).
- Simberloff, D. & Von Holle, B. Positive interactions of nonindigenous species: invasional meltdown? *Biol Invasions* 1, 21-32 (1999).
- Wang, X. et al. Anthropogenic habitat loss accelerates the range expansion of a global invader. *Divers Distrib*, doi: <https://doi.org/10.1111/ddi.13359> (2022).

Thank you very much for your comment.

Reference:

- Blackburn, T. M., Pyšek, P., Bacher, S., Carlton, J. T., Duncan, R. P., Jarošík, V., ... & Richardson, D. M. (2011). A proposed unified framework for biological invasions. *Trends in ecology & evolution*, 26(7), 333-339.
- Ficetola, G. F., Maiorano, L., Falcucci, A., Dendoncker, N., Boitani, L., PADOA-SCHIOPPA, E. M. I. L. I. O., ... & Thuiller, W. (2010). Knowing the past to predict the future: land-use change and the distribution of invasive bullfrogs. *Global change biology*, 16(2), 528-537.
- Hudson, L. N., Newbold, T., Contu, S., Hill, S. L., Lysenko, I., De Palma, A., ... & Milder, J. C. (2014). The PREDICTS database: a global database of how local terrestrial biodiversity responds to human impacts. *Ecology and evolution*, 4(24), 4701-4735.
- Hudson, L. N., Newbold, T., Contu, S., Hill, S. L., Lysenko, I., De Palma, A., ... & Eigenbrod, F. (2017). The database of the PREDICTS (projecting responses of ecological diversity in changing terrestrial systems) project. *Ecology and evolution*, 7(1), 145-188.
- Kalusová, V., Chytrý, M., Peet, R. K., & Wentworth, T. R. (2015). Intercontinental comparison of habitat levels of invasion between temperate North America and Europe. *Ecology*, 96(12), 3363-3373.015, pp. 3363–3373.
- Newbold, T., Hudson, L. N., Hill, S. L., Contu, S., Lysenko, I., Senior, R. A., ... & Purvis, A. (2015). Global effects of land use on local terrestrial biodiversity. *Nature*, 520(7545), 45-50.
- Semenchuk, P., Plutzer, C., Kastner, T., Matej, S., Bidoglio, G., Erb, K. H., ... & Dullinger, S. (2022). Relative effects of land conversion and land-use intensity on terrestrial vertebrate diversity. *Nature communications*, 13(1), 1-10.

Redding, D. W., Pigot, A. L., Dyer, E. E., Şekerciöğlü, Ç. H., Kark, S., & Blackburn, T. M. (2019). Location-level processes drive the establishment of alien bird populations worldwide. *Nature*, 571(7763), 103-106.

Wang, X., Yi, T., Li, W., Xu, C., Wang, S., Wang, Y., ... & Liu, X. (2022). Anthropogenic habitat loss accelerates the range expansion of a global invader. *Diversity and Distributions*, 28(8), 1610-1619.

REVIEWER COMMENTS

Reviewer #1 (Remarks to the Author):

I appreciate all the efforts of the authors to address my comments. I like the study, I think it could be a great contribution to the field. I still have some concerns regarding the global message of the paper and how the spatial coverage might affect the conclusion of the paper.

-I think the abstract reads better, but still it gives the misleading impression that the results are strong among the five taxa, which is not the case.

L74-75: The assemblages of primary vegetation were the least invaded but only for plants if you take into account the statistical analyses. The differences are not significant for ants (secondary), birds (secondary and pasture), mammals (pasture) and spiders (secondary and pasture). Or did I misread the results from Fig 2 Table S8 ? (see Iso L306-308)

My main concern is still about the spatial coverage, the authors compared the results between the different taxa, by assuming that the only differences between taxa assemblages are the taxa considered, while if you check the geographical distribution of local assemblages (Fig S1), there is a strong spatial variability of local assemblages among taxa. As a consequence, I am not sure if the detected differences in the paper are due to the taxa or the spatial distribution of the local assemblages.

Moreover, given most of the results are driven towards Europe and North America for plants at least. The spatial coverage is one of the main limitation that should be discussed. At least, the authors could expand their analyses by taking into account the spatial distribution of local assemblages. Fig S1 showed that there is no European assemblages for ants, but most of the spiders assemblages come from Europe. The spatial coverage of local assemblages is very taxa dependent. The authors described the taxa coverage L143-152, but they did not describe the spatial coverage. I am not sure why it should not be considered at the same level than the taxa analyses; especially with such a strong variability.

L276 bird assemblages least invaded, is it possible that this result is due to the spatial coverage of the local assemblages (How many islands?) In fact, plants are the most invaded assemblages but this also might be due to spatial distributions of plant assemblages (mostly in Europe) while other taxa do not have most of their assemblages in Europe or US, which are known to be the hotspots of invasions.

L329 : the taxa and spatial coverage should be mentioned as one of the limitation that may affect how the invisibility of LU-types may differ among taxonomic groups. I do not think the taxa results could be compared without controlling somehow the spatial distribution of the local assemblages.

Minor comment :

-I think the title reads weird:" The impact of land use on alien species incidence and number in local assemblages worldwide". The number in local assemblages should be presented first and then how land use affects this pattern.

Reviewer #2 (Remarks to the Author):

Comments on revised ms "impact of land use ..."

Comments are as encountered, and not in order of significance.

Assemblages is still problematic for me, as this simply means there was no control for temporal or spatial scale or sampling methods, all of which will have an enormous impact on species lists. The spatial extent of samples varies by orders of magnitude. This will make a difference - - and could also be accounted for by including sampled area in the models.

Models aren't provided, so they can't be assessed. But methods use both frequentist and information theoretic approaches, which are fundamentally different. Better to use one approach. And as stated before, oceanic islands are so unique biologically - and from a NIS perspective, that they should not be mixed with continental faunas. Simply increases variance.

I am still unsatisfied with the term alien - it results from a 1996 US executive order and EU administrative decisions, is not a biological definition, includes economic harm in the definition (which is ignored by the authors), and is a loaded term that should not be used in scientific literature. MS includes other terms such as naturalized. Non-indigenous is much simpler and clearer, and science-based. Authors define alien as introduced, but where - what about naturally invading species, or those intro elsewhere and then spreading to sites?

Spatial extents are now provided as a range, but not incorporated into models, but should be.

Primary veg as "untouched by human actions" - untouched by human action does not exist anywhere on this planet.

State a "global-scale" analysis, but really just a compilation of mostly very local, opportunistic data. Also, disagree its not possible to differentiate intro, establish and spread at a local scale - - and authors argued this is a "global scale" analysis. Scale remains undefined here.

Includes data from same sites with different sampling approaches - these are independent - are they treated as such in models?

Agree data has huge bias in it.

Livestock raising often has relatively little impact, cows often functionally replace native herbivores. Livestock v. rowcrop effects are very different.

Don't think LU "fosters" "naturalization and spread". How, what does this mean?

The world is highly non-stationary, especially now. NIS may functionally replace lost native species. And abundance matters. Richness alone does not convey much information, especially in regards to making sweeping conclusions. The biases discussed could be lessened by more carefully vetting the data, and including variables related to sampling in the models.

Disturbed environments - conveys no info. What does that mean?

Reviewer #3 (Remarks to the Author):

I am overall satisfied with the revisions and the manuscript has been much improved. I suggest that there is still one issue regarding the sampling bias in primary vegetation habitat, which needs to be clarified. I can understand that the authors did not discuss more on the biotic resistance hypothesis and invasion melt-down hypothesis regarding the potential explanation on the observed less

invasion in primary vegetation habitat due to space limitation. However, the authors need address the potential sampling bias issue in primary vegetation habitat. I notice that the authors have discussed the sampling bias issue in the “caveats” section, but in addition to the potentially sampling bias across continents, biomes and taxa, there might also be differences in sampling efforts among habitats with different land-use intensities. Compared with habitats with high human intensity, primary vegetation habitat may be more difficult to be reached by humans due to less accessibility despite that similar sampling protocol might be used in primary and non-primary habitats included in the PREDICTS database.

Responses to the reviewers

Reviewer #1 (Remarks to the Author):

I appreciate all the efforts of the authors to address my comments. I like the study, I think it could be a great contribution to the field. I still have some concerns regarding the global message of the paper and how the spatial coverage might affect the conclusion of the paper.

Thank you very much for your positive comment.

-I think the abstract reads better, but still it gives the misleading impression that the results are strong among the five taxa, which is not the case.

L74-75: The assemblages of primary vegetation were the least invaded but only for plants if you take into account the statistical analyses. The differences are not significant for ants (secondary), birds (secondary and pasture), mammals (pasture) and spiders (secondary and pasture). Or did I misread the results from Fig 2 Table S8 ? (see Iso L306-308)

We partly agree, but would like to point out that for each taxon there is at least one, and mostly several, LU-types that do have a higher incidence of non-native species than in primary vegetation, and, vice versa, for no taxon there is any LU-type which has a significantly lower incidence of non-natives than in primary vegetation. We hence wrote: “Differences to other land-use types were most consistent in case of plants, but for none of the taxa primary vegetation was more invaded than any other land-use type”.

However, to further clarify the difference between the results of plants and the other taxa, we now modified this part to (Line 74-77): *“In plants, assemblages of primary vegetation were least invaded. In the other taxa, primary vegetation was among the least invaded land-use types, but one or several other types had equally low levels of occurrence, frequency and proportions of non-native species.”*

My main concern is still about the spatial coverage, the authors compared the results between the different taxa, by assuming that the only differences between taxa assemblages are the taxa considered, while if you check the geographical distribution of local assemblages (Fig S1), there is a strong spatial variability of local assemblages among taxa. As a consequence, I am not sure if the detected differences in the paper are due to the taxa or the spatial distribution of the local assemblages.

We fully agree that there is considerable spatial heterogeneity in sampling intensity in general, and that there are also pronounced differences among taxa with respect to which regions have been sampled more or less densely. This can lead to falsely attributing differences among regions to taxa. We hence added the following additional analyses to assess the sensitivity of our results to these regional effects:

1. We first used the assignment of each assemblage in PREDICTS to the 14 biomes distinguished by Olson et al. (2001) as an additional random factor in all models (i.e. the random-effects structure was study-site blocks nested in study sites nested in biomes (1|SSB/SS/biome) [or, simpler, (1|SS/biome) in some cases], both for the models fitted across all taxa and for the ones fitted for each taxon separately. We found that including this additional random intercept of biome did not decrease the AIC of the models by more than two points in any of the models (cf. the new Supplementary Table S7). Models with an additional random factor for biome were also not significantly different from those without when compared by

means of a likelihood ratio test (except the one model of non-native occurrence which differed from its counterpart without biome in the random factor with a p-value of 0.048). These findings suggest that accounting for the spatial distribution of PREDICTS assemblages on the globe does not significantly improve our models.

2. We further assessed the robustness of our modelling results against the spatial variation of the assemblages by a leave-one-out cross-validation similar to that in Newbold et al. (2015). We therefore re-ran the full models (including LU-type and -intensity and their interaction) and the LU-type-only model for non-native incidence, numbers and proportion under exclusion of all assemblages from one biome in turn. The random-effects structure was the same as in the original models (study-site blocks nested in study sites (SSB/SS) or study sites (SS)). Subsequently, mean coefficient estimates, standard errors, and 95% confidence intervals of coefficient estimates across all models were calculated using the summarise function in R-package *dplyr* (version 1.0.9). This cross-validation did not change the qualitative patterns and conclusions drawn from our original results (cf. the new Supplementary Table S8 and Figure S5, 6).

We report the results of this sensitivity analysis in the Results section of the paper (Line 189-195 and Line 253-259) and also return to it in the Discussion (Line 415-428). We describe the methods used at the end of the Methods and Materials section (Line 621-639) and provide the respective R-code on github (“R code for testing the robustness of result across biomes.Rmd”, <https://github.com/liudyuk/data-and-code-for-alien-invasion-in-local-assemblage.git>)

Moreover, given most of the results are driven towards Europe and North America for plants at least. The spatial coverage is one of the main limitation that should be discussed. At least, the authors could expand their analyses by taking into account the spatial distribution of local assemblages. Fig S1 showed that there is no European assemblages for ants, but most of the spiders assemblages come from Europe. The spatial coverage of local assemblages is very taxa dependent. The authors described the taxa coverage L143-152, but they did not describe the spatial coverage. I am not sure why it should not be considered at the same level than the taxa analyses; especially with such a strong variability.

This comment relates to the previous one. Again, we agree and point to the sensitivity analyses described in response to the previous comment, which suggest that the spatial distribution of assemblages has little effect on the reported results. In addition, we have expanded the Caveats section to make the still remaining limitations clearer.

L276 bird assemblages least invaded, is it possible that this result is due to the spatial coverage of the local assemblages (How many islands?) In fact, plants are the most invaded assemblages but this also might be due to spatial distributions of plant assemblages (mostly in Europe) while other taxa do not have most of their assemblages in Europe or US, which are known to be the hotspots of invasions.

In fact, birds are a group with a relatively high percentage of assemblages from islands (17% of the total of 4925 assemblages, as compared to 44.4% of total 773 assemblages in Spiders). It is hence very unlikely that they appear least invaded because assemblages are biased against islands in this taxonomic group in particular. Vice versa, it is true that many plant assemblages are from Europe (but not from the US), which might have contributed to the high incidence of

non-native species in this group. We have now included this aspect in the Discussion (see first paragraph of the Caveats section, Line 415-420).

“Moreover, the spatial coverage of different geographical areas and biomes is uneven across taxa (Fig. S1), with large gaps especially in ants (few temperate assemblages) and spiders (few tropical assemblages). Differences in the levels of invasion among groups (Fig. 1) may be related to these spatial biases, because regional pools of non-native species also show pronounced geographical variability²⁸.”

L329 : the taxa and spatial coverage should be mentioned as one of the limitations that may affect how the invisibility of LU-types may differ among taxonomic groups. I do not think the taxa results could be compared without controlling somehow the spatial distribution of the local assemblages.

As explained in our response to the previous comments, we agree on this point and have therefore done additional sensitivity analyses plus an appropriate expansion of the Caveats section (Line 189-195, Line 253-259, Line 415-428 and Line 621-639).

Minor comment:

-I think the title reads weird:” The impact of land use on alien species incidence and number in local assemblages worldwide”. The number in local assemblages should be presented first and then how land use affects this pattern.

As acknowledged elsewhere by the referee the focus of this paper is on the impact of land use on non-native species in local assemblages, not on the number of non-natives in local assemblages per se. We think the title reflects this focus better as it is and would hence prefer to stay with it.

#####

Reviewer #2 (Remarks to the Author):

Comments on revised ms "impact of land use ..."

Comments are as encountered, and not in order of significance.

We thank you for your comments.

Assemblages is still problematic for me, as this simply means there was no control for temporal or spatial scale or sampling methods, all of which will have an enormous impact on species lists. The spatial extent of samples varies by orders of magnitude. This will make a difference - - and could also be accounted for by including sampled area in the models.

We agree and have now included the spatial extent of samples into the models in a sensitivity analysis. We could not do so for the analyses represented in the main text because information on sampling area was not available for all assemblages. However, we refitted the original models for the subset of samples for which we had this information with and without sample size included as an additional fixed-effects predictor. We did not find significant effects of

sample size on non-native incidence, numbers or proportions. The regression coefficients in models with and without sample size included were very similar in all models, and significance only differed in two out of 52 comparisons (all main effects and interactions in all three models of incidence, abundance and proportions). We hence conclude that the results reported are robust against variation in the extent of the sampling area.

We describe the methods of this additional sensitivity analysis in the Materials and Methods section (Line 640-646) and refer to the results in the main text (Line 253-257) and Supplementary Table S9. The R-code used to run the analysis can be found on github (“R code for testing sampling size and island and continent.Rmd”, <https://github.com/liudyuk/data-and-code-for-alien-invasion-in-local-assemblage.git>).

Models aren't provided, so they can't be assessed. But methods use both frequentist and information theoretic approaches, which are fundamentally different. Better to use one approach.

The statistical model results are provided in the supporting tables. The R code for running all the statistical models and making figures are provided on github. (<https://github.com/liudyuk/data-and-code-for-alien-invasion-in-local-assemblage.git>).

We agree that the information-theory approach underlying AIC values, and the ‘frequentist-statistics’-approach underlying p values are based on different concepts. However, we think there is a value in reporting both, especially if they deliver consistent results as is almost always the case in our analyses. We would thus prefer to stay with how we report our results.

And as stated before, oceanic islands are so unique biologically - and from a NIS perspective, that they should not be mixed with continental faunas. Simply increases variance.

We agree that biological invasions on islands are somewhat special. In response to this comment, we have therefore assigned the location of each assemblage to either an island or a continent (Line 647-658). We then re-fitted the models of non-native species incidence, numbers and proportions with this assignment as an additional (nested) random effect. We compared these models with those that did not include the island-continent distinction in the random intercepts by means of AIC (and likelihood ratio tests) and did not find any significant differences (Table S10).

Additionally, we re-ran the models after excluding assemblages from islands and compared the mean estimates and standard errors of the two models. We could not run the models for islands alone because the number of assemblages from islands was not large enough. The results of this additional analysis can be found in Table S11 and Fig S7,S8. Excluding island assemblages led to higher alien incidence in plantations under light use intensity, and lower alien proportions in urban areas under intense use. All other results were almost identical to the ones reported in the main manuscript. We added these two deviations to the Results sections and conclude that our results are largely robust against mixing sites from islands and continents.

Details can be found on github under “R code for testing sampling size and island and continent” (<https://github.com/liudyuk/data-and-code-for-alien-invasion-in-local-assemblage.git>).

I am still unsatisfied with the term alien - it results from a 1996 US executive order and EU administrative decisions, is not a biological definition, includes economic harm in the

definition (which is ignored by the authors), and is a loaded term that should not be used in scientific literature. MS includes other terms such as naturalized. Non-indigenous is much simpler and clearer, and science-based. Authors define alien as introduced, but where - what about naturally invading species, or those intro elsewhere and then spreading to sites?

We understand these terminological reservations and now replaced the term ‘alien’ with “non-native” in the whole manuscript.

Spatial extents are now provided as a range, but not incorporated into models, but should be.

We have now included the spatial extent of the sampling area into our models. Please, see our response to the respective earlier comment for details.

Primary veg as "untouched by human actions" - untouched by human action does not exist anywhere on this planet.

Thanks for pointing us to this error. We have corrected the description which is now also more in line with the original definition in Hudson et al. 2014: *natural habitats not known to have ever been completely destroyed by human actions or natural events that are not part of the natural disturbance regime of the respective ecosystems. Includes synonyms such as “ancient woodlands”, “old-growth forests” or “natural grasslands”*.

State a "global-scale" analysis, but really just a compilation of mostly very local, opportunistic data. Also, disagree its not possible to differentiate intro, establish and spread at a local scale - - and authors argued this is a "global scale" analysis. Scale remains undefined here.

We argue that ‘scale’ has two dimensions, grain and extent. The data in PREDICTS have a local grain, but a global extent, i.e. they include local assemblages from across the globe. We hence think we correctly state that we take a global perspective, and we also make the specific combination of grain and extent clear in the title and in the abstract by combining the keywords ‘local’ and ‘worldwide’. We fully agree, however, that the distribution of assemblages across the globe in the PREDICTS data involves a lot of biases, which we also discuss (and for which we now provide sensitivity analyses, see Figs. S5-8 and Tables S7-11).

With respect to the differentiation of invasion status this is usually done at a regional extent. We could imagine distinguishing introduced and established at a local scale, but doing the same for spread appears impossible as, by definition, it includes the spread between different local assemblages. In any case, there is no such information available for individual assemblages. We emphasize, however, that the regional databases of non-native species include only species classified as naturalized or invasive (= rapidly spreading) at this regional extent (as defined in the Methods section).

Includes data from same sites with different sampling approaches - these are independent - are they treated as such in models?

All statistical analyses are based on mixed-effects models that take account of the spatial structure of sampling by including study sites (and blocks within study sites if appropriate) in the random effects. Within a study site, sampling efforts can usually be considered to be

equal across all plots (according to Hudson et al. 2014; Newbold et al., 2015). Please, see the Data analysis subchapter in the Materials and Methods section for more details.

Agree data has huge bias in it.

Yes. We have now included tests to evaluate the sensitivity of our results to these biases and also expanded the Caveats section of the discussion on this issue.

Livestock raising often has relatively little impact, cows often functionally replace native herbivores. Livestock v. rowcrop effects are very different.

We agree that livestock often replaces shrunken or extinct populations of native herbivores and hence could have less drastic effects on ecosystems than conversion to agricultural fields. However, there are also many instances where forests have been removed (or are still removed) to create grasslands for livestock grazing, with massive implications for species composition and diversity.

Don't think LU "fosters" "naturalization and spread". How, what does this mean?

Thank you, we have replaced "foster" by "promote".

The world is highly non-stationary, especially now. NIS may functionally replace lost native species. And abundance matters. Richness alone does not convey much information, especially in regards to making sweeping conclusions. The biases discussed could be lessened by more carefully vetting the data, and including variables related to sampling in the models.

We agree that not all non-native species cause problems and that some of them may even have beneficial effects and we have integrated this aspect into the Conclusions section. We also agree that analysing data on non-native abundances would deliver important further insights. The biases discussed are now included in additional sensitivity analyses as far as possible - please, compare the last part of the Materials and Methods section (Line 621-658), the Figures S5-S8, the Tables S7-S11, and our responses to the related comments of the referee above.

Disturbed environments - conveys no info. What does that mean?

Replaced by "disturbed conditions"

#####

Reviewer #3 (Remarks to the Author):

I am overall satisfied with the revisions and the manuscript has been much improved.

Thank you very much for your comment.

I suggest that there is still one issue regarding the sampling bias in primary vegetation habitat, which needs to be clarified. I can understand that the authors did not discuss more on the biotic resistance hypothesis and invasion melt-down hypothesis regarding the potential

explanation on the observed less invasion in primary vegetation habitat due to space limitation. However, the authors need address the potential sampling bias issue in primary vegetation habitat. I notice that the authors have discussed the sampling bias issue in the “caveats” section, but in addition to the potentially sampling bias across continents, biomes and taxa, there might also be differences in sampling efforts among habitats with different land-use intensities. Compared with habitats with high human intensity, primary vegetation habitat may be more difficult to be reached by humans due to less accessibility despite that similar sampling protocol might be used in primary and non-primary habitats included in the PREDICTS database.

We agree and have included this aspect now into the Caveats section of the Discussion (Line 432-439).

“On the other hand, assemblages from the vast cold and arid regions of the globe are also poorly represented, and these are often sparsely invaded¹⁴. Further, if non-native species are much rarer than native species in a study site, they will be more likely to have been missed or misidentified during sampling, as they are not expected to be present there. Sampling efforts in the original studies underlying PREDICTS may have also differed across LU-types, taxa and biomes³. For instance, rare non-native species may have been difficult to detect in remote primary vegetation with low accessibility, even if sampling protocols were standardized within individual studies²⁶.”

References:

- Hudson, L. N. et al. The PREDICTS database: A global database of how local terrestrial biodiversity responds to human impacts. *Ecol Evol* 4, 4701–4735 (2014).
- Newbold, T. et al. Global effects of land use on local terrestrial biodiversity. *Nature* **520**, 45–50 (2015).
- Olson, D. M. et al. Terrestrial Ecoregions of the World: A New Map of Life on Earth: A new global map of terrestrial ecoregions provides an innovative tool for conserving biodiversity. *BioScience*, **51**, 933–938 (2001).

REVIEWERS' COMMENTS

Reviewer #1 (Remarks to the Author):

I would like to thank the authors for taking into account previous rounds of comments. I think the revised version reads better and is now much stronger than previous one. This paper will make an important contribution to the field. I still have some minor comments to consider for improvements.

The sensitivity analyses to take into account the role of biomes and location of the assemblage (island vs. mainland) is important but to help the readers to understand the results, the significant differences with letters should be added to Fig S5, S6, S7 and S8. It is very difficult to compare the results using the figures from the main text, the supplementary figures and the tables.

L297-301: The authors made a rough comparison and add new results by comparing the % of invaded assemblages in their data with Van Kleunen paper. I understand why the authors made that comparison but this is very simplified and this number could be used out of context without understanding the limitations of the datasets and the oversimplifications. I believe the paper is already strong and do not need this comparison, which add new results in the discussion that are not properly presented in the main result.

L350: The authors mentioned several reasons to explain the invisibility of LU-types, but they did not mention the intrinsic mobility of plants compared to other organisms. I was wondering if this could also explain their results, given the limited mobility of plants (particularly on islands) compared to birds, mammals, ants, etc.

Reviewer #3 (Remarks to the Author):

The authors have provided more discussions on the sampling bias issue in their data used. I don't have other specific concerns on this manuscript.

REVIEWERS' COMMENTS

Reviewer #1 (Remarks to the Author):

I would like to thank the authors for taking into account previous rounds of comments. I think the revised version reads better and is now much stronger than previous one. This paper will make an important contribution to the field.

Thank you.

I still have some minor comments to consider for improvements.

The sensitivity analyses to take into account the role of biomes and location of the assemblage (island vs. mainland) is important but to help the readers to understand the results, the significant differences with letters should be added to Fig S5, S6, S7 and S8. It is very difficult to compare the results using the figures from the main text, the supplementary figures and the tables.

We do not test the different significances by statistical analyses. We conducted the sensitivity analyses to assess the robustness of the modelling results by a leave-one-out cross-validation. And then mean coefficient estimates, standard errors, and 95% confidence intervals of coefficient estimates across all models were calculated using the summarise function in R-package dplyr (version 1.0.9). We showed the cross-validated coefficient estimates, standard errors and confidence intervals in these figures (FigS5-S8) and the tables (Table S7-S11), which can be used to compare the values and conclusions drawn from our original results.

L297-301: The authors made a rough comparison and add new results by comparing the % of invaded assemblages in their data with Van Kleunen paper. I understand why the authors made that comparison but this is very simplified and this number could be used out of context without understanding the limitations of the datasets and the oversimplifications. I believe the paper is already strong and do not need this comparison, which add new results in the discussion that are not properly presented in the main result.

We agree and have made changes accordingly. As this argument does not only concern the calculation for plants in the lines 304-313, but also the ones for the other groups in the following lines, we have removed them altogether and replaced them by a similar, but more generic argumentation without any numbers. The new text reads as follows:

“However, the total number of species naturalized outside their native range is much higher in plants than in the other groups (Supplementary Table 2). A higher incidence of non-native plants in local assemblages would thus also result from neutral community assembly processes. Given that there are almost 50 times more non-native plants than non-native birds, mammals, spider or ants (Supplementary Table 2), the incidence of vascular plants might even appear low, and those of some other groups high in relative terms. However, whether there really are non-random differences in non-native incidence among taxonomic would need a more thorough analysis. The PREDICTS data were not designed and are hence difficult to use for such an analysis, mainly because of a number of taxonomic, environmental and geographical sampling biases. “

L350: The authors mentioned several reasons to explain the invisibility of LU-types, but they did not mention the intrinsic mobility of plants compared to other organisms. I was wondering if this could also explain their results, given the limited mobility of plants (particularly on islands) compared to birds, mammals, ants, etc.

We agree again and have added this argument to this paragraph of the Discussion (Lines 362-368):

“Third, the taxonomic groups considered differ in their intrinsic mobility, with plants probably being least mobile³¹. Given that primary vegetation is often the LU-type most distant from points of non-native species introduction, plants may simply be slower in colonizing primary vegetation than species from the other groups. Our results would thus document differences in the level by which species from different taxonomic groups have already realized their potential ranges in the non-native regions.”

Reviewer #3 (Remarks to the Author):

The authors have provided more discussions on the sampling bias issue in their data used. I don't have other specific concerns on this manuscript.

Thank you.